# FAITHFUL RULE EXTRACTION FOR DIFFERENTIABLE RULE LEARNING MODELS

**Xiaxia Wang, David J. Tena Cucala, Bernardo Cuenca Grau, Ian Horrocks**
Department of Computer Science, University of Oxford, UK
`{xiaxia.wang,david.tenacucala,bernardo.grau,ian.horrocks}@cs.ox.ac.uk`

## ABSTRACT

There is increasing interest in methods for extracting interpretable rules from ML models trained to solve a wide range of tasks over knowledge graphs (KGs), such as KG completion, node classification, question answering and recommendation. Many such approaches, however, lack formal guarantees establishing the precise relationship between the model and the extracted rules, and this lack of assurance becomes especially problematic when the extracted rules are applied in safety-critical contexts or to ensure compliance with legal requirements. Recent research has examined whether the rules derived from the influential NEURAL-LP model exhibit soundness (or completeness), which means that the results obtained by applying the model to any dataset always contain (or are contained in) the results obtained by applying the rules to the same dataset. In this paper, we extend this analysis to the context of DRUM, an approach that has demonstrated superior practical performance. After observing that the rules currently extracted from a DRUM model can be unsound and/or incomplete, we propose a novel algorithm where the output rules, expressed in an extension of Datalog, ensure both soundness and completeness. This algorithm, however, can be inefficient in practice and hence we propose additional constraints to DRUM models facilitating rule extraction, albeit at the expense of reduced expressive power.

## 1 INTRODUCTION

Numerous tasks over knowledge graphs (KGs) such as completion (Wang et al., 2021), node classification (Portisch & Paulheim, 2022), question answering (Lan et al., 2021) and recommendation (Guo et al., 2022; Xian et al., 2019) can be formulated as a transformation from an input to an output dataset (e.g., KG completion transforms an incomplete KG to its extension with the missing facts). ML models are widely employed to acquire these transformations from examples, as it is a more cost-effective approach than manual design and does not require domain expertise.

ML-based solutions face a challenge in explaining predictions. Recent studies (Yang et al., 2017; Evans & Grefenstette, 2018; Sadeghian et al., 2019; Qu et al., 2021; Ferreira et al., 2022; Zhang et al., 2023; Wang et al., 2023) propose techniques to extract interpretable Datalog (Abiteboul et al., 1995) rules from trained models. However, many of these approaches lack formal guarantees establishing the relationship between the model and the extracted rules. Instead, they often rely on informal claims that the rules "approximate" or "explain" the model's behavior (Yang et al., 2017; Evans & Grefenstette, 2018; Sadeghian et al., 2019; Qu et al., 2021). Such claims may be substantiated by empirical evidence showcasing the similarity between the model's predictions and the outcomes of rule application (Ferreira et al., 2022). Nonetheless, the need for formal assurances regarding the alignment between model and rules becomes particularly critical when these rules are applied in safety-critical contexts or to ensure compliance with legal requirements for explainability.

Recent studies have explored the formal relationship between models and extracted rules in the context of Feed-Forward Networks (Ayoobi et al., 2023; Wang et al., 2022), Convolutional Neural Networks (Benamira et al., 2023), and Graph Neural Networks (Tena Cucala et al., 2022a;b). These works examine whether the rules exhibit *soundness* (or *completeness*), which means that the results obtained by applying the model to any given dataset always contain (or are contained in) the results obtained by applying the rules to the same dataset. *Faithful* (i.e., both sound and complete) rule sets

can be used to explain the model's predictions. For example, assume that a model $\mathcal{M}_{med}$ trained to suggest potential diagnoses predicts fact diagnose$(alice, flu)$ when applied to the dataset

$$\mathcal{D}_{med} = \{\text{indicativeOf}(fever, flu), \text{indicativeOf}(fever, tuberculosis),$$
$$\text{hasSymptom}(alice, fever), \text{contact}(alice, bob), \text{diagnose}(bob, flu)\}.$$

A medical professional can examine the extracted rules $\mathcal{R}_{med}$ to understand model predictions. An incomplete rule set may fail to support the predicted fact, leaving the prediction unexplained. An unsound rule, like diagnose$(x, y) \leftarrow$ hasSymptom$(x, z) \wedge$ indicativeOf$(z, y)$, may generate facts not predicted by the model, such as diagnose$(alice, tuberculosis)$. In contrast, a faithful rule set lacks spurious rules and guarantees an explanatory rule for each predicted fact. For example, the following rule $r_{diag}$ explains $alice$'s flu diagnosis due to her fever and contact with a flu patient:

$$\text{diagnose}(x, y) \leftarrow \text{hasSymptom}(x, z_1) \wedge \text{indicativeOf}(z_1, y) \wedge \text{contact}(x, z_2) \wedge \text{diagnose}(z_2, y).$$

The analysis in Tena Cucala et al. (2022b) revealed that soundness for NEURAL-LP can be ensured by selecting appropriate hyperparameters, but completeness cannot be guaranteed. Thus, the extracted rules may not be a faithful representation of the model.

In this paper, we study faithfulness guarantees in the context of DRUM (Sadeghian et al., 2019)—an approach inspired by NEURAL-LP that has demonstrated superior empirical performance. DRUM exhibits significant differences with respect to NEURAL-LP, which makes the relationship between these approaches unclear. First, each DRUM model comprises multiple sub-models, initially applied independently to the input data, followed by the aggregation of their outputs; each DRUM sub-model resembles a NEURAL-LP model, but it contains much fewer parameters as it does not implement "skip connections" between network layers. Second, DRUM sub-models and NEURAL-LP employ different mechanisms for learning rules of varying lengths. Third, DRUM can generate a broader class of rules, including inverse rules such as parent$(x, y) \leftarrow$ child$(y, x)$.[1] Given these disparities, the results from Tena Cucala et al. (2022b) for NEURAL-LP do not directly apply to DRUM.

In Section 2, we revisit the definitions of Datalog, DRUM, and the concepts of soundness, completeness, and faithfulness introduced in Tena Cucala et al. (2022b). In Section 3, we analyse the faithfulness of rules extracted from a DRUM model. We observe that, much like in NEURAL-LP, the behavior of DRUM can be characterised in terms of "counting" the distinct matches of a rule's body within the input data. We can thus establish results akin to those in Tena Cucala et al. (2022b) regarding NEURAL-LP. In Section 4, we present a method for extracting a faithful set of rules from a DRUM model expressed in an extension of Datalog with inequalities and disjunctions in the rule body, enabling the necessary counting operations. This represents an advancement compared to prior research because the faithful rule extraction method in Tena Cucala et al. (2022b) applies only to significantly restricted NEURAL-LP models. However, we note that achieving a practical implementation of our method may be challenging due to the time complexity associated with the rule extraction algorithm. In Section 5 we propose two solutions to this issue. Firstly, we introduce a mechanism that, given a DRUM model and a dataset, extracts a rule set that is sound for the model and derives all the model predictions on the given dataset. Secondly, we impose constraints on the DRUM models facilitating the extraction of a faithful rule set, at the expense of reduced expressive power. Specifically, we introduce two variants of the model, MMDRUM and SMDRUM, each striking a distinct balance between expressivity of the model and the effectiveness of rule extraction.

In Section 6, we conduct a comprehensive evaluation on KG completion tasks. Amongst other findings, our experiments show that SMDRUM and MMDRUM obtain competitive performance and confirm the practical feasibility of the rule extraction algorithms proposed in Section 5.

## 2  BACKGROUND

**Datalog.**   A *signature* consists of disjoint, countable sets of *constants* and *predicates*, where each predicate is assigned a non-negative *arity*. A *term* is a *variable* or a constant. An *atom* is an expression of the form $R(t_1, \cdots, t_n)$, where $R$ is an $n$-ary predicate and $t_1, \cdots, t_n$ are terms. A *fact* is a variable-free atom, and a *dataset* is a finite set of facts. A *Datalog rule* is an expression of the form:

$$H \leftarrow B_1 \wedge \cdots \wedge B_\ell, \quad \ell \geq 0, \tag{1}$$

---

[1] There is an additional difference concerning the way in which the model's parameters are generated; however, it is irrelevant to our analysis, which does not rely on the method used to compute the model's parameters.

where $H$ and $B_i$ for $1 \leq i \leq \ell$ are atoms. Typically, $H$ is called the *head atom*, and each $B_i$ is called a *body atom*. The value $\ell \in \mathbb{N}$ is the *length* of the rule. We do not make the usual *safety* requirement that each variable in $H$ must appear in some $B_i$. In fact, the rule body can be empty if $\ell = 0$, in which case we write $\top$.[2] A *Datalog program* is a finite set of rules.

For a mapping $\sigma$ of variables to constants and a term or a conjunction $X$ with variables in the domain of $\sigma$, $X\sigma$ is the result of replacing each $x$ in $X$ with $\sigma(x)$. For a rule $r$ of the form (1), the *immediate consequence* operator $T_r$ maps a dataset $\mathcal{D}$ to the smallest dataset $T_r(\mathcal{D})$ containing $H\sigma$ for each mapping $\sigma$ of variables in $r$ to constants in $\mathcal{D}$ that *grounds* the body of $r$ in $\mathcal{D}$ i.e. that satisfies $B_i\sigma \in \mathcal{D}$ for all $1 \leq i \leq \ell$. For a Datalog program $\mathcal{R}$, $T_{\mathcal{R}}$ is defined as $T_{\mathcal{R}}(\mathcal{D}) = \bigcup_{r \in \mathcal{R}} T_r(\mathcal{D})$.

**Vectors, Matrices and Tensors.** For each $n \in \mathbb{N}$, we use the notion of a $n$-dimensional *tensor* (Yang et al., 2017; Sadeghian et al., 2019) over $\mathbb{R}$ (e.g., a *matrix* is a 2-tensor). For an $n$-tensor $\mathbf{A}$, $\mathbf{A}(i_1, i_2, \cdots, i_n)$ is the element at position $(i_1, i_2, \cdots, i_n)$. If $\mathbf{M}$ and $\mathbf{N}$ are matrices of dimension $m \times n$ and $n \times p$, respectively, then the *max-product* of $\mathbf{M}$ and $\mathbf{N}$, written $\mathbf{M} \otimes \mathbf{N}$, is a matrix of dimension $m \times p$ whose element at position $i$ and $j$ is equal to $\max_{k=1}^{n} \mathbf{M}(i, k) \cdot \mathbf{N}(k, j)$.

**The DRUM Model.** DRUM (Sadeghian et al., 2019) assumes a signature with $\delta$ binary predicates $R_1, R_2, \cdots, R_\delta$ and is designed to learn rules of the form

$$R_h(x, y) \leftarrow \psi_{k_1} \wedge \psi_{k_2} \wedge \cdots \wedge \psi_{k_\ell} \text{ with } \ell \geq 1 \text{ and } h, k_i \in \{1, \cdots, \delta\}, \tag{2}$$

where $\psi_{k_i}$ is of the form $R_{k_i}(z_{i-1}, z_i)$ or $R_{k_i}(z_i, z_{i-1})$, with $z_0 = x$ and $z_\ell = y$. We will show in Section 3, however, that rules of this form are not sufficiently expressive to faithfully characterise DRUM models. For instance, our example rule $r_{diag}$ in Section 1 is not of this form.

A DRUM model $\mathcal{M}$ of rank $N \geq 1$ and depth $L \geq 1$ is a tuple $(\mathbf{a}^1, \cdots, \mathbf{a}^\delta, \beta)$, where each $\mathbf{a}^h$ for $1 \leq h \leq \delta$ is a 3-tensor over $[0, 1]^{N \times L \times (2\delta+1)}$ of (learnable) parameters and $\beta \in \mathbb{R}$ is a *threshold* for prediction. The rank determines the number of considered "sub-models", as mentioned in Section 1. Models with higher rank are more expressive and can simulate a richer set of rules (Sadeghian et al., 2019). The depth $L$ determines the maximum length of a rule that can be learned. A DRUM model $\mathcal{M}$ induces a transformation $T_{\mathcal{M}}$ over datasets as we specify next. Given a dataset $\mathcal{D}$, let $c_1, c_2, \cdots, c_\epsilon$ be the constants in $\mathcal{D}$ arranged in a fixed but arbitrary order. For each $1 \leq k \leq 2\delta + 1$, an $\epsilon \times \epsilon$ binary matrix $\mathbf{M}_k$ is computed by setting to 1 all values $\mathbf{M}_k(i, j)$ satisfying one of the following conditions: *(1)* $R_k(c_i, c_j) \in \mathcal{D}$ and $1 \leq k \leq \delta$, *(2)* $R_{k-\delta}(c_j, c_i) \in \mathcal{D}$ and $\delta + 1 \leq k \leq 2\delta$, or *(3)* $i = j$ and $k = 2\delta + 1$; all remaining values are set to 0.

Matrices $\mathbf{M}_k$ for $k \leq \delta$ are adjacency matrices representing facts in $\mathcal{D}$. DRUM can be conceptualised as extending the input dataset with new facts over $\delta + 1$ fresh predicates in the remaining matrices: a fact $R_{2\delta+1}(c, c)$ for each constant $c$ in $\mathcal{D}$ and a fact $R_{\delta+k}(c_j, c_i)$ for each $R_k(c_i, c_j) \in \mathcal{D}$ representing its *inverse*. The adjacency matrices $\mathbf{M}_{\delta+1}, \cdots, \mathbf{M}_{2\delta+1}$ capture the new facts. Next, DRUM represents each constant $c_s$ with an one-hot vector $\mathbf{v}_s \in \{0, 1\}^\epsilon$ in which $\mathbf{v}_s(s) = 1$ and 0 elsewhere. Then, for $1 \leq h \leq \delta$, DRUM computes a vector $\mathbf{v}_s^h$ as Equation (3).

$$(\mathbf{v}_s^h)^\intercal = (\mathbf{v}_s)^\intercal \cdot \sum_{i=1}^{N} \prod_{j=1}^{L} \left( \sum_{k=1}^{2\delta+1} \mathbf{a}^h(i, j, k) \cdot \mathbf{M}_k \right). \tag{3}$$

Vector $\mathbf{v}_s^h$ indicates, for each $1 \leq t \leq \epsilon$, whether the fact $R_h(c_s, c_t)$ is in $T_{\mathcal{M}}(\mathcal{D})$, in the following way: $R_h(c_s, c_t) \in T_{\mathcal{M}}(\mathcal{D})$ if and only if $\mathbf{v}_s^h(t)$ is strictly greater than the prediction threshold $\beta$.

**Rule Extraction in DRUM.** Rules are extracted from a trained DRUM model $\mathcal{M}$ of depth $L$ by first computing a confidence score $\alpha_r \in \mathbb{R}$ for each rule $r$ of the form (2) with length at most $L$. We next show how to compute such score. In the simple case where $r$ has length exactly $L$, the score is $\alpha_r = \sum_{i=1}^{N} \alpha_r^i$ where each $\alpha_r^i$ is simply the product of the model parameters corresponding to each body atom of $r$ for the $i$-th sub-model. In particular, for $R_h$ the head predicate of $r$, parameter $\mathbf{a}^h(i, j, k_j)$ (respectively, $\mathbf{a}^h(i, j, k_j + \delta)$) is associated to a body atom $R_{k_j}(z_{j-1}, z_j)$ (respectively, $R_{k_j}(z_j, z_{j-1})$ at position $1 \leq j \leq L$ and sub-model $1 \leq i \leq N$. Finally, if the length of $r$ is smaller than $L$, we first consider all the ways in which $r$ can be extended to an equivalent

---

[2]Rules are applied to datasets with finitely many constants, and thus only finitely many facts can be derived.

rule of the form (2) and length exactly $L$ by padding the sequence of predicates in the body with the identity predicate $R_{2\delta+1}$. For example, if $L = 2$, a rule $R_2(x, y) \leftarrow R_1(x, y)$ can be extended as

$$R_2(x, y) \leftarrow R_1(x, z_1) \land R_{2\delta+1}(z_1, y) \quad \text{and} \quad R_2(x, y) \leftarrow R_{2\delta+1}(x, z_1) \land R_1(z_1, y).$$

We can compute a value for each of these rules as before and set $\alpha_r$ as the maximum of these values.

For a rule $r$ of the form (2), the set $\mathcal{S}_r^L$ contains all sequences $(k_1', \cdots, k_L')$ that can be obtained from $(k_1, \cdots, k_\ell)$ by replacing $k_j$ by $k_j + \delta$ if $\psi_{k_j}$ is of the form $R_{k_j}(z_j, z_{j-1})$, and then padding it (if needed) with the value $2\delta + 1$. Then, the confidence score $\alpha_r$ for $r$ is

$$\alpha_r = \max_{(k_1', \cdots, k_L') \in \mathcal{S}_r^L} \sum_{i=1}^{N} \prod_{j=1}^{L} \mathbf{a}^h(i, j, k_j'). \tag{4}$$

For each $\gamma \in \mathbb{R}$, $\mathcal{R}_{\mathcal{M}, \gamma}^{\text{DRUM}}$ is the set of all rules in the form (2) with $\ell \leq L$ and score higher than $\gamma$.

**Relations between Models & Programs.** Program $\mathcal{R}$ is *sound* for model $\mathcal{M}$ if $T_{\mathcal{R}}(\mathcal{D}) \subseteq T_{\mathcal{M}}(\mathcal{D})$ for any dataset $\mathcal{D}$. Conversely, $\mathcal{R}$ is *complete* for $\mathcal{M}$ if $T_{\mathcal{M}}(\mathcal{D}) \subseteq T_{\mathcal{R}}(\mathcal{D})$ for any dataset $\mathcal{D}$. Finally, we say that $\mathcal{R}$ is *faithful* for $\mathcal{M}$ if it is both sound and complete for $\mathcal{M}$ (Tena Cucala et al., 2022b).

## 3 ANALYSING FAITHFULNESS OF RULE EXTRACTION IN DRUM

We begin by describing the behavior of a DRUM model $\mathcal{M}$ of rank $N$ and depth $L$. We show that when $\mathcal{M}$ is applied to a dataset $\mathcal{D}$, the decision made by $\mathcal{M}$ as to whether to return a fact $R_h(c_s, c_t)$ can be characterised by considering each rule of the form (2) with $\ell \leq L$ and head predicate $R_h$, and then counting the distinct matches of the rule body in $\mathcal{D}$ where $x$ is mapped to $c_s$ and $y$ to $c_t$.

**Lemma 1.** *For a dataset $\mathcal{D}$ with constants $c_1, \cdots, c_\epsilon$, vector $\mathbf{v}_s^h$ computed by $\mathcal{M}$ for $1 \leq s \leq \epsilon$ and $1 \leq h \leq \delta$ is equal to $\sum_{r \in \mathcal{R}_{h,L}^{\text{PATH}}} \varphi_{\mathcal{M}}(r) \cdot \mathbf{q}_{r, \mathcal{D}, s}$, where $\mathcal{R}_{h,L}^{\text{PATH}}$ is the set of all rules of the form (2) with $1 \leq \ell \leq L$ (resp. $\ell = 0$) and head atom $R_h(x, y)$ (resp. $R_h(x, x)$), $\varphi_{\mathcal{M}}(r)$ is a non-negative function of $r$ that depends only on the parameters of $\mathcal{M}$, and $\mathbf{q}_{r, \mathcal{D}, s}$ is a vector of dimension $\epsilon$ such that, its $t$-th element is 1 (resp. 0) if the body of $r$ is $\top$ and $t = s$ (resp. $t \neq s$), and otherwise it is the number of different mappings that ground the body of $r$ in $\mathcal{D}$ and map $x$ to $c_s$ and $y$ to $c_t$.*

Consider a DRUM model with $N = 1$, $L = 2$, $\beta = 1.5$, $\mathbf{a}^2(1, 1, 1) = 1$, $\mathbf{a}^2(1, 2, 2) = 1$, and all other tensor elements equal to 0. Suppose *contact* and *diagnose* are the first and second predicates in the signature, respectively, and *alice* and *flu* are in positions $s$ and $t$ of the constant order, respectively. Equation (3) ensures that the $t$-th element of $\mathbf{v}_s^2$ represents the count of distinct constants $c$ where both contact(*alice*, $c$) and diagnose($c$, *flu*) are in the dataset. The value of $\beta$ ensures that diagnose(*alice*, *flu*) is derived if and only if *alice* has been in contact with at least two flu patients, indicating at least two valid paths from *alice* to *flu*.

In (3), the right-hand side computes a product of the one-hot vector $\mathbf{v}_s$ and $L$ adjacency matrices for extended dataset $\mathcal{D}'$ with inverse and self-reflective facts. Each vector element corresponds to a constant $c_t$ and represents the count of paths within $\mathcal{D}'$ of length $L$ from $c_s$ to $c_t$. These paths traverse edges labeled with predicates corresponding to the adjacency matrices, following the specified sequence. Each path corresponds uniquely to a subset of $\mathcal{D}$ grounding of the body of a rule (2) and is associated with a weight derived from products and summations of elements of the tensor $\mathbf{a}^h$. The aggregation of such weights yields the expression in Lemma 1 showing that DRUM shares similarities with NEURAL-LP, where predictions depend also on the counts of rule matches. The approaches used by NEURAL-LP and DRUM to compute rule weights, however, differ significantly and it is unclear whether NEURAL-LP models can be simulated by DRUM models.

The aforementioned similarities allow us to derive results analogous to those obtained for NEURAL-LP concerning the connection between the model and the rules extracted from it.

**Theorem 1.** *Program $\mathcal{R}_{\mathcal{M}, \gamma}^{\text{DRUM}}$ is sound for $\mathcal{M} = (\mathbf{a}^1, \cdots, \mathbf{a}^\delta, \beta)$ whenever $\gamma \geq \beta$. Furthermore, there is a DRUM model $\mathcal{M}' = (\mathbf{a}'^1, \cdots, \mathbf{a}'^\delta, \beta')$ such that $\mathcal{R}_{\mathcal{M}', \gamma}^{\text{DRUM}}$ is unsound for $\mathcal{M}'$ for any $\gamma < \beta'$.*

Hence, by selecting a suitable value for $\gamma$, it is possible to ensure the soundness of the extracted rules. Nevertheless, achieving completeness cannot be guaranteed.

**Theorem 2.** *There exists a DRUM model such that no Datalog program is faithful for it.*

## 4 FAITHFUL RULE EXTRACTION FOR DRUM

Theorem 2 shows that DRUM models cannot be fully expressed using Datalog. This is unsurprising considering that DRUM models are characterised in terms of the counts of unique matches of rule bodies in datasets deriving relevant predictions. In contrast, Datalog rules derive a fact whenever there exists a match of the rule body in the data, regardless of the count of distinct relevant matches. Datalog can, however, be extended with counting (Dantsin et al., 2001) by introducing inequality atoms in rule bodies interpreted under the *Unique Name Assumption (UNA)*, which stipulates that any two distinct constants must refer to separate entities. For instance, consider the following rules:

$$R_2(x,y) \quad \leftarrow \quad R_1(x,z_1^1) \wedge R_1(z_1^1,y) \wedge R_1(x,z_1^2) \wedge R_1(z_1^2,y) \,. \tag{5}$$

$$R_2(x,y) \quad \leftarrow \quad R_1(x,z_1^1) \wedge R_1(z_1^1,y) \wedge R_1(x,z_1^2) \wedge R_1(z_1^2,y) \wedge z_1^1 \not\approx z_1^2 \,. \tag{6}$$

On dataset $\mathcal{D}_1 = \{R_1(c_1,c_2), R_1(c_2,c_3)\}$, rule (5) derives fact $R_2(c_1,c_3)$ through a match assigning $x \mapsto c_1$, $y \mapsto c_3$, $z_1^1 \mapsto c_2$ and $z_1^2 \mapsto c_2$. In contrast, rule (6) does not apply to $\mathcal{D}_1$ since variables $z_1^1$ and $z_1^2$ cannot be mapped to the same constant. On dataset $\mathcal{D}_2 = \mathcal{D}_1 \cup \{R_1(c_1,c_4), R_1(c_4,c_3)\}$, however, both rules derive $R_2(c_1,c_3)$. In particular, rule (6) admits the match $x \mapsto c_1$, $y \mapsto c_3$, $z_1^1 \mapsto c_2$ and $z_1^1 \mapsto c_4$ given that the inequality $c_2 \not\approx c_4$ holds by the UNA.

In this section, we demonstrate that such extended Datalog programs can faithfully represent DRUM models. Additionally, we introduce disjunction in the rule bodies: this is a convenient extension that can (exponentially) reduce the number of rules required without increasing the expressive power of the language. Indeed, each rule with disjunction in the body is equivalent to multiple disjunction-free rules. For example, the rule $R_3(x,y) \leftarrow (R_1(x,z_1) \vee R_2(x,z_1)) \wedge (R_1(z_1,y) \vee R_2(z_1,y))$ is equivalent to the four rules of the form $R_3(x,y) \leftarrow R_{k_1}(x,z_1) \wedge R_{k_2}(z_1,y)$, with $\{k_1,k_2\} \subseteq \{1,2\}$.

The basic building block in rule bodies is a *multipath conjunction* specified by a *cardinality* $C \in \mathbb{N}$ and a *core* $\Psi$ with the same structure as the rule body in Equation (2). Intuitively, a multipath conjunction consists of $C$ copies of its core where variables other than $x, y$ have been renamed; each such copy represents a "path". Meanwhile, inequalities are incorporated to guarantee that no two paths can be matched to the dataset in an identical manner. Thus, a multipath conjunction is satisfied if it admits at least $C$ distinct matches of its core agreeing on the assignments of variables $x$ and $y$.

**Definition 1.** *Let $\Psi$ be a conjunction of length $\ell \geq 1$ in the form of the rule body in* (2)*. A* multipath conjunction $\phi$ *with* core $\Psi$ *and* cardinality $C \in \mathbb{N}$ *is of the form*

$$\phi = \bigwedge_{j=1}^{C} \Psi^j \ \wedge \bigwedge_{1 \leq j < j' \leq C} \left( \bigvee_{i=1}^{\ell-1} z_i^j \not\approx z_i^{j'} \right) , \tag{7}$$

*where $\Psi^j$ replaces in $\Psi$ each $z_i$ by $z_i^j$ for $1 \leq i \leq \ell-1$. Its length is $\ell$. A* multipath rule *of length $L$ has head $R_h(x,y)$ (resp. $R_h(x,x)$) and its body is a conjunction of $P \geq 1$ (resp. $P \geq 0$ and where $x = y$) multipath conjunctions of length $\leq L$ and pairwise disjoint variables other than $x$ and $y$.*

The definition of the immediate consequence operator is standard and given by the well-known semantics of Datalog with inequalities and disjunction (Dantsin et al., 2001) under the UNA.

Counting in multipath rules is limited by the cardinalities of their multipath conjunctions. In contrast, DRUM models face no such restrictions as the relevant counts are data-dependent and can become arbitrarily large. To address this challenge, we turn to Lemma 1 where $\mathcal{M}$ assigns a "weight" $\varphi_{\mathcal{M}}(r)$ to each rule of the form (2). There is no need to count the matches of a rule $r$ with weight zero, as the resulting value will be multiplied by zero. Similarly, when there is at most one body atom, a single match is possible. For positive weights and rules with multiple body atoms, there is a minimum number of matches ensuring that the corresponding value in vector $\mathbf{v}_s^h$ exceeds the model's prediction threshold $\beta$, regardless of the other rules. The threshold will be met if the number of matches $\omega(r)$ for $r$ reaches $\lfloor \frac{\beta}{\varphi_{\mathcal{M}}(r)} \rfloor + 1$. Thus, we can ignore rules featuring multipath conjunctions $\phi_p$ with cardinalities exceeding $\omega(R_h(x,y) \leftarrow \Psi_p)$, where $\Psi_p$ is the core of $\phi_p$; any such rule would be equivalent to the rule obtained by substituting $C_p$ with $\omega(R_h(x,y) \leftarrow \Psi_p)$. This leads to a finite set of relevant multipath rules, and the only remaining challenge lies in constructing the specific subset characterising the model.

Algorithm 1 outlines the procedure for extracting a faithful program from a DRUM model. It begins by initializing the output program as an empty set (line 1). Additionally, it creates a list $\Omega$ encompassing all conjunctions serving as the body of rules in the form of (2) with length $\ell \leq L$, followed

---

**Algorithm 1:** Multipath Rule Extraction.

---

**Input:** A DRUM model $\mathcal{M} = (\mathbf{a}^1, \cdots, \mathbf{a}^\delta, \beta)$, and a rule extraction threshold $\gamma$.
**Output:** A multipath program

1   $\mathcal{R} := \emptyset$;
2   $\Omega :=$ list of all conjunctions in the form of the body of (2) with $0 \leq \ell \leq L$, ending with $\top$;
3   **foreach** $h \in \{1, \cdots, \delta\}$ **do**
4     **foreach** $[k_1, \cdots, k_L]$ with $k_i \in \{1, \cdots, 2\delta + 1\}$ **do**
5       $[k'_1, \cdots, k'_\ell] :=$ remove all $2\delta + 1$;
6       **foreach** $j \in \{1, \cdots, \ell\}$ **do**
7         **if** $k'_j \leq \delta$ **then** $\psi_j := R_{k'_j}(z_{j-1}, z_j)$; **else** $\psi_j := R_{k'_j - \delta}(z_j, z_{j-1})$;
8       **if** $\ell \geq 1$ **then** $r := R_h(x, y) \leftarrow \bigwedge_{j=1}^{\ell} \psi_j$; **else** $r := R_h(x, x) \leftarrow \top$;
9       **if** $\varphi_r$ is undefined **then** $\varphi_r := 0$;
10      $\varphi_r := \varphi_r + \sum_{i=1}^{N} \prod_{j=1}^{L} \mathbf{a}^h(i, j, k_j)$;
11     **foreach** $i \in \{1, \cdots, |\Omega| - 1\}$ **do**
12       $r_i := R_h(x, y) \leftarrow \Omega(i)$;
13       **if** $\Omega(i)$ has one atom **then** $\omega_i := 1$; **else** $\omega_i := \lfloor \frac{\beta}{\varphi_{r_i}} \rfloor + 1$;
14     **foreach** $[C_1, \cdots, C_{|\Omega|-1}]$ with $C_i \in \{0, \cdots, \omega_i\}$ **do**
15       **if** $C_i = 0$ **then** $\rho_i := \top$ ; **else** $\rho_i :=$ multip. conj. of core $\Omega(i)$ and cardinality $C_i$;
16       **if** $\sum_{i=1}^{|\Omega|-1} C_i \cdot \varphi_{r_i} > \gamma$ **then** $\mathcal{R} := \mathcal{R} \cup \{R_h(x, y) \leftarrow \bigwedge_{i=1}^{|\Omega|-1} \rho_i\}$;
17       **if** $\sum_{i=1}^{|\Omega|-1} C_i \cdot \varphi_{r_i} + \varphi_{r_{|\Omega|}} > \gamma$ **then** $\mathcal{R} := \mathcal{R} \cup \{R_h(x, x) \leftarrow \bigwedge_{i=1}^{|\Omega|-1} \rho_i \{y \mapsto x\}\}$;
18   **return** $\mathcal{R}$;

---

by $\top$ (line 2). All elements of $\Omega$ except the last are possible cores of multipath conjunctions. The subsequent steps of the algorithm involve iterating over all predicates $R_h$ in the signature (line 3). Within each iteration, it adds necessary multipath rules with head atom $R_h(x, y)$ and $R_h(x, x)$ to the output program. Specifically, it first computes the function $\varphi_{\mathcal{M}}$ from Lemma 1 for each rule $r$ as $R_h(x, y) \leftarrow \Psi$ with $\Psi$ in $\Omega$ and $R_h(x, x) \leftarrow \top$, with the results stored in $\varphi_r$ (lines 4–10). Following this, for each rule $r$, the algorithm computes an upper bound $\omega$ on the cardinality of multipath conjunctions with core $\Psi$ (lines 11–13). Subsequently, it enumerates all rules with head predicate $R_h$ where the cardinality of each multipath conjunction does not exceed the computed bound for its core. This enumeration is performed by considering all combinations of cardinalities (line 14) for each core (i.e., each element of $\Omega$ except $\top$) and constructing the corresponding rules for each combination (line 15). The algorithm calculates a score for each rule by summing the products of each multipath conjunction's cardinality with the weight assigned by $\varphi_{\mathcal{M}}$ to its core. The score is compared to a threshold $\gamma \in \mathbb{R}$ and the rule is added if the threshold is exceeded (lines 16–17).

The following theorem establishes the correctness and complexity of our rule extraction algorithm.

**Theorem 3.** *Program $\mathcal{R}_{\mathcal{M},\gamma}^{\mathrm{MP}}$ extracted by Algorithm 1 on input $\mathcal{M} = (\mathbf{a}^1, \cdots, \mathbf{a}^\delta, \beta)$ is faithful to $\mathcal{M}$ for $\gamma = \beta$. Furthermore, Algorithm 1 terminates in $\mathcal{O}\left(L \cdot (2\delta)^{L+1} \left(N + \omega^{(2\delta)^{L+1}}\right)\right)$ steps, where $\omega$ is the maximum value of $\omega(r)$ for $r$ a rule of the form (2) with $\ell \leq L$.*

In the example from Section 3, the extracted program would consist of the following rule of the form (7) with core $\mathrm{contact}(x, z) \wedge \mathrm{diagnose}(z, y)$ and cardinality 2:

$$\mathrm{diagnose}(x, y) \leftarrow \mathrm{contact}(x, z_1) \wedge \mathrm{diagnose}(z_1, y) \wedge \mathrm{contact}(x, z_2) \wedge \mathrm{diagnose}(z_2, y) \wedge z_1 \not\approx z_2 \, .$$

## 5   PRACTICAL RULE EXTRACTION

The complexity of Algorithm 1 underscores the difficulty of extracting faithful programs from DRUM models. We next propose two approaches for addressing this challenge.

### 5.1   RULE EXTRACTION FOR A FIXED DATASET

Our first solution involves extracting a (usually small) subset $\mathcal{R}_{\mathcal{M}}^{\mathcal{D}}$ of sound rules for $\mathcal{M}$ that explain all the model's predictions on a given dataset $\mathcal{D}$. Focusing on a concrete dataset suffices in scenarios where data is not subject to frequent updates, and makes rule extraction practically feasible.

---

**Algorithm 2:** Multipath Rule Extraction for a Fixed Dataset.

**Input:** A DRUM model $\mathcal{M} = (\mathbf{a}^1, \cdots, \mathbf{a}^\delta, \beta)$, and a dataset $\mathcal{D}$.
**Output:** A multipath program.

1   $\mathcal{R} := \emptyset$;
2   $\mathcal{P}_{\text{next}} := \{[c_s] \mid R_h(c_s, c_t) \in T_{\mathcal{M}}(\mathcal{D})\}, \mathcal{P}_{\text{all}} := \mathcal{P}_{\text{next}}$;
3   **foreach** $j \in [1, \cdots, L]$ **do**
4        $\mathcal{P}_{\text{current}} := \mathcal{P}_{\text{next}}$;
5        $\mathcal{P}_{\text{next}} := \emptyset$ ;
6        **while** $\mathcal{P}_{\text{current}}$ *is not empty* **do**
7            **pop** $[\cdots, c_{s'}]$ **from** $\mathcal{P}_{\text{current}}$ ;
8            **foreach** $R_k(c_{s'}, c_{t'}) \in \mathcal{D}$ **do** $\mathcal{P}_{\text{next}} := \mathcal{P}_{\text{next}} \cup \{[\cdots, c_{s'}, k, c_{t'}]\}$;
9            **foreach** $R_k(c_{t'}, c_{s'}) \in \mathcal{D}$ **do** $\mathcal{P}_{\text{next}} := \mathcal{P}_{\text{next}} \cup \{[\cdots, c_{s'}, k + \delta, c_{t'}]\}$;
10        $\mathcal{P}_{\text{all}} := \mathcal{P}_{\text{all}} \cup \mathcal{P}_{\text{next}}$;
11  **foreach** $R_h(c_s, c_t) \in T_{\mathcal{M}}(\mathcal{D})$ **do**
12       $\rho := \top$;
13       $\texttt{count} : \emptyset \mapsto 0$;
14       **foreach** $[c_s, k_1, \cdots, k_\ell, c_t] \in \mathcal{P}_{\text{all}}$ **do**
15           **foreach** $j \in \{1, \cdots, \ell\}$ **if** $k_j \leq \delta$ **then** $\psi_j := R_{k_j}(z_{j-1}, z_j)$; **else** $R_{k'_j - \delta}(z_j, z_{j-1})$;
16           $\texttt{count}(\bigwedge_{j=1}^{\ell} \psi_j) := \texttt{count}(\bigwedge_{j=1}^{\ell} \psi_j) + 1$;
17       **foreach** $\psi : \texttt{count}(\psi) > 0$ **do**
18           append to $\rho$ a multipath conj. of core $\psi$ and cardinality $\min(\texttt{count}(\psi), \omega(R_h(x, y) \leftarrow \psi))$;
19       **if** $s \neq t$ **then** $\mathcal{R} := \mathcal{R} \cup \{R_h(x, y) \leftarrow \rho\}$; **else** $\mathcal{R} := \mathcal{R} \cup \{R_h(x, x) \leftarrow \rho\{y \mapsto x\}\}$;
20  **return** $\mathcal{R}$;

---

Algorithm 2 implements this idea. Given a model $\mathcal{M}$ and a dataset $\mathcal{D}$, the first part of the algorithm applies $\mathcal{M}$ to $\mathcal{D}$ and computes all "paths" $\mathcal{P}_{\text{all}}$ of length at most $L$ in $\mathcal{D}$ that start from a constant $c_s$ featuring in a fact of the form $R_h(c_s, c_t) \in T_{\mathcal{M}}(\mathcal{D})$. The set $\mathcal{P}_{\text{all}}$ is initialised with paths of length 0 (i.e., all relevant constants). Subsequently, for each $1 \leq j \leq L$, it iteratively determines the set of all paths of length $j$ (lines 4–10). This is achieved by considering all paths with a length of $j-1$ (line 7) and examining all possible extensions they can undergo (lines 8–9). The second part of the algorithm adds a rule $r$ to $\mathcal{R}_{\mathcal{M}}^{\mathcal{D}}$ for each fact $R_h(c_s, c_t)$ in $T_{\mathcal{M}}(\mathcal{D})$ so that the application of $r$ to $\mathcal{D}$ derives this fact. Rule $r$ is initialised with an empty body (line 12). Then, the algorithm counts all paths in $\mathcal{P}_{\text{all}}$ that start in $c_s$, end in $c_t$ (line 14), and traverse the same sequence of predicates (line 15–16), and adds a corresponding multipath conjunction to the body of $r$ (lines 17–19).

**Theorem 4.** *Program $\mathcal{R}_{\mathcal{M}}^{\mathcal{D}}$ extracted by Algorithm 2 for a DRUM model $\mathcal{M} = (\mathbf{a}^1, \cdots, \mathbf{a}^\delta, \beta)$ and a dataset $\mathcal{D}$ satisfies $T_{\mathcal{R}_{\mathcal{M}}^{\mathcal{D}}}(\mathcal{D}) = T_{\mathcal{M}}(\mathcal{D})$. Furthermore, Algorithm 2 terminates with time complexity $\mathcal{O}((2\delta)^L \cdot (N \cdot L + 2\delta \cdot \epsilon^{L+2}))$.*

The complexity of Algorithm 2 depends on the structure of $\mathcal{D}$ and $T_{\mathcal{M}}(\mathcal{D})$. If their adjacency matrices are dense, the algorithm will exhibit exponential behaviour. In practice, however, the matrices are usually sparse (see Table 4). As shown in Section 6, program $\mathcal{R}_{\mathcal{M}}^{\mathcal{D}}$ can be efficiently computed.

### 5.2 FAITHFUL RULE EXTRACTION BY LIMITING MODEL EXPRESSIVITY

Our second approach involves simplifying DRUM models so that extracting faithful programs from them becomes practically feasible, at the expense of decreasing the expressive power of the model.

A key source of complexity in Algorithm 1 is the need to enumerate all relevant multipath rules. In particular, the enumeration of all possible combinations of cardinalities causes an exponential blowup. This can be avoided by sacrificing the model's capacity to count distinct rule matches in the data. Instead, it goes back to Datalog, retaining only the capability to check whether the body atoms of a rule can be matched or not. This simplification can be achieved by pushing the vector $(\mathbf{v}_s)^\intercal$ inside the product and sum operators in Equation (3)—which yields an equivalent expression—and then replacing the sum over $k$ with a max aggregation, and matrix products with matrix max-products. The resulting expression cannot be written in a compact form as in Equation (3), but we can write it concisely by using an inductive definition. We call the resulting model SMDRUM.

**Definition 2.** *An* SMDRUM *model is defined as a* DRUM *model, but Equation* (3) *is replaced by* $(\mathbf{v}_s^h)^\intercal = \sum_{i=1}^{N} \mathbf{u}_i^L$, *where* $\mathbf{u}_i^0 = (\mathbf{v}_s)^\intercal$, *and* $\mathbf{u}_i^j = \max_{1 \leq k \leq 2\delta+1} \mathbf{a}^h(i,j,k) \cdot \mathbf{u}_i^{j-1} \otimes \mathbf{M}_k$ *for* $1 \leq j \leq L$.

This restriction limits expressivity: each SMDRUM model is equivalent to a set of multipath rules where each rule has up to $N$ multipath conjunctions without inequalities (i.e., of cardinality 1). Such rules, however, are still more expressive than those of form (2); for instance, they include rule $r_{diag}$ from Section 1.They can be extracted with a simplified version of Algorithm 1 in exponential time.

**Proposition 1.** *For an* SMDRUM *model* $\mathcal{M}$ *there exists a program* $\mathcal{R}_{\mathcal{M}}^{\mathrm{SM}}$ *of inequality-free multipath rules computable in exponential time in* $\mathcal{M}$'s *size such that* $T_{\mathcal{M}}(\mathcal{D}) = T_{\mathcal{R}_{\mathcal{M}}^{\mathrm{SM}}}(\mathcal{D})$ *for any dataset* $\mathcal{D}$.

Unfortunately, extracting a faithful program from an SMDRUM model remains challenging due to the large number of distinct combinations of $N$ multipath conjunctions. To address this issue, we limit model expressivity further by replacing the sum over $1 \leq i \leq N$ by another max aggregation.

**Definition 3.** *An* MMDRUM *model is defined as an* SMDRUM *model, but where the sum over* $1 \leq i \leq N$ *is replaced by a* max *function.*

The advantage of MMDRUM is that we can extract a faithful program by simply enumerating all rules of the form (2), then computing the score for each rule using a variant of expression (4),

$$\alpha_{\mathcal{M},r} = \max_{(k_1', \cdots, k_L') \in \mathcal{S}_r^L} \max_{1 \leq i \leq N} \prod_{j=1}^{L} \mathbf{a}^h(i,j,k_j'), \tag{8}$$

and then outputting those rules with score higher than a threshold $\gamma \in \mathbb{R}$. This can be achieved efficiently by taking advantage of structure-sharing between rules (see Appendix B). The downside is the further loss in expressivity, as we no longer can express commonly used rules with multiple atoms involving variables $x$ and $y$ in the body e.g., citizenOf$(x, y) \leftarrow$ bornIn$(x, y) \wedge$ livesIn$(x, y)$.

**Theorem 5.** *For an* MMDRUM *model* $\mathcal{M}$ *of depth* $L$, *the program containing each rule* $r$ *of the form* (2) *with* $\ell \leq L$ *or* $R(x, x) \leftarrow \top$ *such that* $\alpha_{\mathcal{M},r} > \beta$ *is faithful for* $\mathcal{M}$.

## 6 EVALUATION

We evaluated MMDRUM, SMDRUM, DRUM, and NEURAL-LP (as a baseline) on the inductive KG completion task. Besides, we also applied and evaluated our rule extraction methods introduced in Section 5. All experiments were conducted on a Linux workstation with a Xeon E5-2670 CPU.

We followed the methodology of Yang et al. (2017) to evaluate DRUM models on KG completion tasks. We considered an *inductive* setting, where constants seen at test time may not have encountered during training. This is in contrast with the *transductive* setting, where all relevant constants occur already in the train set (Bordes et al., 2013; Sun et al., 2019).

We used the 13 benchmark datasets (Appendix C) for inductive KG completion by Teru et al. (2020) based on FB15k-237 (Toutanova & Chen, 2015), NELL-995 (Xiong et al., 2017), WN18RR Dettmers et al. (2018), and Family (Kok & Domingos, 2007), preserving the splits for train, validation and test. We selected $L = 2$ for all models and rank $N = 3$ for all DRUM-based models. Each model was trained up to 10 epochs. Threshold $\beta \in (0, 1)$ for each model is a hyperparameter. We tried several values and picked the one maximising F1-score on validation sets.

We evaluated each model using precision, recall, accuracy, area under the precision-recall curve (AUPRC), and F1 score as metrics. The results are presented in Table 1. Notably, DRUM outperformed both MMDRUM and SMDRUM in terms of overall performance, demonstrating the ability to exploit higher expressive power in practice. For most FB15k-237 and NELL-995 datasets, the four models exhibited comparable performances, while for WN18RR datasets we see significant disparities. In particular, NEURAL-LP showed lower recall than all DRUM-based models, suggesting that the use of multiple "sub-models" ($N > 1$) can be critical in practice, even for MMDRUM.

We also implemented the rule extraction algorithms in Section 5 and applied them to the inductive KG completion benchmarks. In all cases, both Algorithm 2 and the algorithm for MMDRUM (Appendix B) succeeded to compute the corresponding rule sets in less than 3 minutes, which suggest

Table 1: Results (%) of DRUM (D), SMDRUM (S) MMDRUM (M) and NEURAL-LP (N).

| | | Precision | | | | Recall | | | | Accuracy | | | | AUPRC | | | | F1 Score | | | |
|---|---|---|---|---|---|---|---|---|---|---|---|---|---|---|---|---|---|---|---|---|---|
| | | M | S | D | N | M | S | D | N | M | S | D | N | M | S | D | N | M | S | D | N |
| FB15k-237 | V1 | 47.2 | 47.2 | 48.2 | **81.9** | **44.9** | **44.9** | 43.9 | 28.8 | 47.3 | 47.3 | 48.3 | **61.2** | 49.6 | 48.4 | **51.6** | 50.0 | **46.0** | **46.0** | 45.9 | 42.6 |
| | V2 | 53.8 | 53.9 | 58.9 | **89.3** | **56.3** | **56.3** | 54.6 | 33.3 | 54.0 | 54.1 | 58.3 | **64.6** | 60.5 | 60.2 | **63.3** | 60.8 | 55.0 | 55.1 | **56.7** | 48.5 |
| | V3 | 55.7 | 55.6 | 61.0 | **80.6** | 50.1 | **50.4** | 48.9 | 41.8 | 55.1 | 55.1 | 58.8 | **65.9** | 58.7 | 58.0 | **60.6** | 59.4 | 52.7 | 52.9 | 54.3 | **55.1** |
| | V4 | 59.5 | 59.5 | 72.6 | **92.7** | **55.0** | 54.1 | 49.2 | 40.9 | 58.7 | 58.6 | 65.3 | **68.9** | 62.8 | 62.3 | **65.1** | 64.7 | 57.1 | 56.6 | **58.7** | 56.8 |
| NELL-995 | V1 | **100** | 97.2 | 97.4 | **100** | 19.0 | 70.0 | **76.0** | 13.0 | 59.5 | 84.0 | **87.0** | 56.5 | 66.9 | 77.5 | **84.4** | 65.9 | 31.9 | 81.4 | **85.4** | 23.0 |
| | V2 | 64.0 | 64.1 | 64.5 | **66.3** | 75.4 | 75.0 | **75.2** | 71.6 | 66.5 | 66.5 | 66.9 | **67.6** | 69.7 | 71.6 | **74.6** | 70.8 | 69.2 | 69.1 | **69.5** | 68.9 |
| | V3 | 62.2 | 62.1 | **62.6** | 62.5 | **79.4** | 79.0 | 78.5 | 77.1 | 65.6 | 65.4 | **65.8** | 65.4 | 77.2 | 75.6 | **82.0** | 76.0 | **69.7** | 69.5 | 69.6 | 69.1 |
| | V4 | 66.4 | 66.4 | 66.4 | **81.4** | **77.6** | **77.6** | **77.6** | 59.9 | 69.2 | 69.2 | 69.2 | **73.1** | 74.3 | 76.3 | **81.0** | 71.5 | **71.6** | **71.6** | **71.6** | 69.0 |
| WN18RR | V1 | 68.2 | 79.1 | **93.6** | 86.7 | **68.6** | 64.4 | 62.2 | 20.7 | 68.4 | 73.7 | **79.0** | 58.8 | 68.1 | 72.6 | **75.0** | 65.5 | 68.4 | 71.0 | **74.8** | 33.5 |
| | V2 | 57.3 | 76.0 | **100** | 77.8 | **67.1** | 62.4 | 60.8 | 15.9 | 58.5 | 71.3 | **80.4** | 55.7 | 60.4 | 67.7 | **73.4** | 59.1 | 61.8 | 68.5 | **75.6** | 26.4 |
| | V3 | 37.0 | 50.0 | **99.4** | 90.0 | **29.9** | 29.8 | 27.8 | 8.93 | 39.5 | 50.0 | **63.8** | 54.0 | 30.8 | 34.5 | **40.4** | 34.7 | 33.1 | 37.3 | **43.4** | 16.2 |
| | V4 | 68.1 | 78.0 | **94.2** | 87.6 | 59.3 | 59.2 | **59.5** | 9.38 | 65.8 | 71.3 | **77.9** | 54.0 | 62.1 | 66.7 | **70.1** | 61.2 | 63.4 | 67.3 | **72.9** | 16.9 |
| Family | | 93.3 | 92.7 | **97.0** | 94.9 | **92.9** | 87.7 | 82.9 | 92.8 | 93.1 | 90.4 | 90.2 | **93.9** | **94.4** | 94.1 | 94.3 | **94.4** | 93.1 | 90.1 | 89.4 | **93.8** |

Table 3: Rules extracted from Family dataset.

Table 2: Rule Extraction Completeness (%) with $\gamma = \beta$.

| | $\beta$ | 0.0001 | 0.001 | 0.01 | 0.1 |
|---|---|---|---|---|---|
| FB15k-237 | V1 | 3.05 | 3.07 | 0.00 | 0.00 |
| | V2 | 6.16 | 5.84 | 0.00 | 0.00 |
| | V3 | 5.06 | 3.65 | 6.25 | 6.25 |
| | V4 | 5.02 | 5.23 | 3.92 | 5.88 |
| NELL-995 | V1 | 4.12 | 6.19 | 6.13 | 5.17 |
| | V2 | 5.93 | 6.19 | 6.25 | 0.00 |
| | V3 | 5.82 | 6.14 | 6.14 | 3.12 |
| | V4 | 5.65 | 4.05 | 5.74 | 1.70 |
| WN18RR | V1 | 5.88 | 5.28 | 5.74 | 3.76 |
| | V2 | 5.81 | 5.74 | 4.62 | 5.88 |
| | V3 | 6.13 | 5.88 | 4.91 | 4.86 |
| | V4 | 5.81 | 5.62 | 4.51 | 4.78 |

| Top 10 rules extracted by MMDRUM | In Top 10 rules of SMDRUM |
|---|---|
| brother$(x, y) \leftarrow$ brother$(x, z) \land$ brother$(z, y)$ | ✓ |
| uncle$(x, y) \leftarrow$ brother$(x, z) \land$ uncle$(z, y)$ | ✓ |
| sister$(x, y) \leftarrow$ daughter$(x, z) \land$ mother$(z, y)$ | |
| daughter$(x, y) \leftarrow$ sister$(x, z) \land$ daughter$(z, y)$ | ✓ |
| aunt$(x, y) \leftarrow$ sister$(x, z) \land$ aunt$(z, y)$ | ✓ |
| nephew$(x, y) \leftarrow$ nephew$(x, z) \land$ brother$(z, y)$ | |
| nephew$(x, y) \leftarrow$ son$(x, z) \land$ brother$(z, y)$ | |
| sister$(x, y) \leftarrow$ sister$(x, z) \land$ sister$(z, y)$ | ✓ |
| son$(x, y) \leftarrow$ brother$(x, z) \land$ daughter$(z, y)$ | |
| niece$(x, y) \leftarrow$ daughter$(x, z) \land$ brother$(z, y)$ | |
| **Top 5 rules w. different head atoms by SMDRUM that not captured by MMDRUM** | |
| sister$(x, y) \leftarrow$ sister$(x, z_1) \land$ sister$(z_1, y) \land$ sister$(x, z_2) \land$ brother$(y, z_2)$ | |
| niece$(x, y) \leftarrow$ sister$(x, z_1) \land$ niece$(z_1, y) \land$ sister$(x, z_2) \land$ uncle$(y, z_2)$ | |
| brother$(x, y) \leftarrow$ brother$(x, z_1) \land$ brother$(z_1, y) \land$ brother$(x, z_2) \land$ sister$(z_2, y)$ | |
| son$(x, y) \leftarrow$ brother$(x, z_1) \land$ father$(y, z_1) \land$ brother$(x, z_2) \land$ son$(z_2, y)$ | |
| nephew$(x, y) \leftarrow$ brother$(x, z_1) \land$ nephew$(z_1, y) \land$ brother$(x, z_2) \land$ aunt$(y, z_2)$ | |

the practical feasibility of our approach. Additionally, we verified empirically the theoretical guarantees for these algorithms provided in Theorem 4 and Theorem 5.

We implemented the incomplete Datalog rule extraction algorithm for DRUM described in Section 2. We computed the proportion of model predictions covered by the extracted rules for different prediction thresholds. The results in Table 2 show that the rules derive less than 7% of the facts predicted by the models, which suggests that they are insufficient to explain the predictions of the models.

Finally, we examined the extracted rules on the Family dataset for MMDRUM and SMDRUM (the algorithm for SMDRUM is run best effort for a fixed length of time). Table 3 depicts the rules extracted from MMDRUM with the highest score. Some of these were also extracted for SMDRUM and the rankings obtained for both models are similar. We also show the highest ranked rules extracted by SMDRUM not of the form (2). Many of these are in practice irrelevant and suggest overfitting; this makes sense since the performance of MMDRUM in this dataset is close to that of SMDRUM.

## 7 LIMITATIONS AND FUTURE WORK

Rules generated by our approach may be challenging to interpret, as they contain many body atoms. Furthermore, obtaining a faithful program with inequalities for a DRUM model using Algorithm 1 can be challenging in practice. Our first priority will be to devise optimisations for Algorithm 1 that can significantly prune the search space of rules. We also aim at enhancing interpretability by devising algorithms that prioritise shorter rules. Finally, we will extend our analyses to models such as $\partial$ILP (Evans & Grefenstette, 2018), and explore generalisations of DRUM to capture rules with higher-arity predicates.

## REPRODUCIBILITY STATEMENT

The proofs of all the lemmas, theorems, and propositions in this paper are provided in Appendix A. The datasets and source codes used in our experiments are available from the GitHub repository with documentation at `https://github.com/xiaxia-wang/FaithfulRE`.

## ACKNOWLEDGMENTS

This work was supported by the SIRIUS Centre for Scalable Data Access (Research Council of Norway, project 237889), Samsung Research UK, the EPSRC projects AnaLOG (EP/P025943/1), OASIS (EP/S032347/1), UKFIRES (EP/S019111/1), ConCur (EP/V050869/1), and the AIDA project (Alan Turing Institute).

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

## A   PROOFS

**Lemma 1.** *For a dataset $\mathcal{D}$ with constants $c_1, \cdots, c_\epsilon$, vector $\mathbf{v}_s^h$ computed by $\mathcal{M}$ for $1 \le s \le \epsilon$ and $1 \le h \le \delta$ is equal to $\sum_{r \in \mathcal{R}_{h,L}^{\text{PATH}}} \varphi_{\mathcal{M}}(r) \cdot \mathbf{q}_{r,\mathcal{D},s}$, where $\mathcal{R}_{h,L}^{\text{PATH}}$ is the set of all rules of the form (2) with $1 \le \ell \le L$ (resp. $\ell = 0$) and head atom $R_h(x, y)$ (resp. $R_h(x, x)$), $\varphi_{\mathcal{M}}(r)$ is a non-negative function of $r$ that depends only on the parameters of $\mathcal{M}$, and $\mathbf{q}_{r,\mathcal{D},s}$ is a vector of dimension $\epsilon$ such that, its $t$-th element is 1 (resp. 0) if the body of $r$ is $\top$ and $t = s$ (resp. $t \ne s$), and otherwise it is the number of different mappings that ground the body of $r$ in $\mathcal{D}$ and map $x$ to $c_s$ and $y$ to $c_t$.*

*Proof.* Let $\mathcal{D}$ be an arbitrary dataset with constants $c_1, \cdots, c_\epsilon$. The distributive and associative properties of the product and the sum allow us to rewrite Equation (3) as

$$(\mathbf{v}_s^h)^{\mathsf{T}} = \sum_{(k_1, \cdots, k_L) \in \{1, \cdots, 2\delta+1\}^L} \left( \left( \sum_{i=1}^{N} \prod_{j=1}^{L} \mathbf{a}^h(i, j, k_j) \right) (\mathbf{v}_s)^{\mathsf{T}} \cdot \mathbf{M}_{k_1} \cdot \mathbf{M}_{k_2} \cdot \cdots \cdot \mathbf{M}_{k_L} \right). \quad (9)$$

It is straightforward to see that each sequence $(k_1, \cdots, k_L)$ of elements in $\{1, \cdots, 2\delta + 1\}$ corresponds to a rule of the form (2), namely, the unique rule with head atom $R_h(x, y)$ ($R_h(x, x)$ if all elements are $2\delta + 1$) and such that $(k_1, \cdots, k_L) \in S_r^L$. Furthermore, all sequences in $S_r^L$ are over $\{1, \cdots, 2\delta + 1\}^L$ and of length $L$. Hence, we can rewrite Equation (9) as

$$(\mathbf{v}_s^h)^{\mathsf{T}} = \sum_{r \in \mathcal{R}_{h,L}^{\text{PATH}}} \left( \sum_{(k_1, \cdots, k_L) \in \mathcal{S}_r^L} \left( \left( \sum_{i=1}^{N} \prod_{j=1}^{L} \mathbf{a}^h(i, j, k_j) \right) (\mathbf{v}_s)^{\mathsf{T}} \cdot \mathbf{M}_{k_1} \cdot \mathbf{M}_{k_2} \cdot \cdots \cdot \mathbf{M}_{k_L} \right) \right). \quad (10)$$

Let $\mathcal{D}'$ be the extension of $\mathcal{D}$ with all inverse and identity facts over the extended signature, as described in Section 2. A simple inductive argument shows that the vector $(\mathbf{v}_s)^{\mathsf{T}} \cdot \mathbf{M}_{k_1} \cdot \mathbf{M}_{k_2} \cdot \cdots \cdot \mathbf{M}_{k_L}$ describes the number of paths of length $L$ in $\mathcal{D}'$ from $c_s$ to each constant. In particular, the $t$-th element of the vector is the number of paths of length $L$ from $c_s$ to $c_t$ in $\mathcal{D}'$. Furthermore, for each rule $r$ in $\mathcal{R}_{h,L}^{\text{PATH}}$ with $\ell \ge 1$ and for each $(k_1, \cdots, k_L) \in \mathcal{S}_r^L$, there is a one-to-one correspondence between each substitution that grounds the body of $r$ mapping $x$ to $c_s$ and $y$ to $c_t$, and each path of length $L$ from $c_s$ to $c_t$ in $\mathcal{D}'$ through $k_1, k_2, \cdots, k_L$; therefore, for each $(k_1, \cdots, k_L) \in \mathcal{S}_r^L$, the vector $(\mathbf{v}_s)^{\mathsf{T}} \cdot \mathbf{M}_{k_1} \cdot \mathbf{M}_{k_2} \cdot \cdots \cdot \mathbf{M}_{k_L}$ is always $\mathbf{q}_{r,\mathcal{D},s}$. Finally, for $r = R_h(x, x) \leftarrow \top$, there is a unique sequence in $\mathcal{S}_r^L$, namely, that where all elements are $2\delta + 1$. For this sequence, the vector $(\mathbf{v}_s)^{\mathsf{T}} \cdot \mathbf{M}_{k_1} \cdot \mathbf{M}_{k_2} \cdot \cdots \cdot \mathbf{M}_{k_L}$ has $s$-th component equal to 1 and all other components equal to 0, so it is also equal to $\mathbf{q}_{r,\mathcal{D},s}$. Hence, in Equation (10) we can factor out $(\mathbf{v}_s)^{\mathsf{T}} \cdot \mathbf{M}_{k_1} \cdot \mathbf{M}_{k_2} \cdot \cdots \cdot \mathbf{M}_{k_L}$, as it is equal for each $(k_1, \cdots, k_L) \in \mathcal{S}_r^L$ for a given $r$, and replace it by its value $\mathbf{q}_{r,\mathcal{D},s}$. Then the right side of the equation becomes $\sum_{r \in \mathcal{R}_{h,L}^{\text{PATH}}} \varphi_{\mathcal{M}}(r) \cdot \mathbf{q}_{r,\mathcal{D},s}$, as we wanted to prove, with

$$\varphi_{\mathcal{M}}(r) = \sum_{(k_1, \cdots, k_L) \in \mathcal{S}_r^L} \sum_{i=1}^{N} \prod_{j=1}^{L} \mathbf{a}^h(i, j, k_j). \quad (11)$$

$\square$

**Theorem 1.** *Program $\mathcal{R}_{\mathcal{M},\gamma}^{\text{DRUM}}$ is sound for $\mathcal{M} = (\mathbf{a}^1, \cdots, \mathbf{a}^\delta, \beta)$ whenever $\gamma \ge \beta$. Furthermore, there is a DRUM model $\mathcal{M}' = (\mathbf{a}'^1, \cdots, \mathbf{a}'^\delta, \beta')$ such that $\mathcal{R}_{\mathcal{M}',\gamma}^{\text{DRUM}}$ is unsound for $\mathcal{M}'$ for any $\gamma < \beta'$.*

*Proof.* For the first claim, we consider an arbitrary fact $R_h(c_s, c_t) \in T_{\mathcal{R}_{\mathcal{M},\gamma}^{\text{DRUM}}}(\mathcal{D})$, and we show that $R_h(c_s, c_t) \in T_{\mathcal{M}}(\mathcal{D})$. Since $R_h(c_s, c_t) \in T_{\mathcal{R}_{\mathcal{M},\gamma}^{\text{DRUM}}}(\mathcal{D})$, $\mathcal{R}_{\mathcal{M},\gamma}^{\text{DRUM}}$ contains a rule of the form (2) where $0 \le \ell \le L$, and there is a substitution $\sigma$ such that $x\sigma = c_s$, $y\sigma = c_t$, and the grounding of the rule's body by this substitution is in $\mathcal{D}$. Now, by hypothesis, $\gamma \ge \beta$. Moreover, the value $\alpha_r$ from Equation (4) is strictly greater than $\gamma$ because $r \in \mathcal{R}_{\mathcal{M},\gamma}^{\text{DRUM}}$. Furthermore, $\varphi_{\mathcal{M}}(r) \ge \alpha_r$, as seen from comparing Equation (11) and Equation (4). Finally, we have $\mathbf{v}_s^h(t) \ge \varphi_{\mathcal{M}}(r)$ from Lemma 1 and the fact that $\mathbf{q}_{r,\mathcal{D},s}(t) \ge 1$ due to the existence of $\sigma$. Combining all these inequalities yields $\mathbf{v}_s^h(t) > \beta$, indicating $R_h(c_s, c_t) \in T_{\mathcal{M}}(\mathcal{D})$, as we intended to show.

We prove the second claim with an example. Let $\mathcal{D} = \{R_1(c_1, c_1), R_2(c_2, c_2)\}$ be a dataset, and $\mathcal{M} = (\mathbf{a}^1, \cdots, \mathbf{a}^\delta, \beta)$ be a DRUM model in which $N = L = 1$, and $\beta = 0.5$. All non-zero elements of $\mathbf{a}^1, \cdots, \mathbf{a}^\delta$ are $\mathbf{a}^2(1, 1, 1) = \mathbf{a}^2(1, 1, 2) = 0.5$. According to Equation (3), the

model does not predict the fact $R_2(c_1, c_1)$ since $\mathbf{v}_s^h(1) \not\succ \beta$. However, for any $\gamma < \beta$, the program $\mathcal{R}_{\mathcal{M},\gamma}^{\text{DRUM}}$ contains the rule $R_2(x, y) \leftarrow R_1(x, y)$, and $T_{\mathcal{R}_{\mathcal{M},\gamma}^{\text{DRUM}}}(\mathcal{D})$ contains the fact $R_2(c_1, c_1)$. Therefore, $\mathcal{R}_{\mathcal{M},\gamma}^{\text{DRUM}}$ is not sound for $\mathcal{M}$. $\qquad\square$

**Theorem 2.** *There exists a* DRUM *model such that no Datalog program is faithful for it.*

*Proof.* Let $\mathcal{M} = (\mathbf{a}^1, \cdots, \mathbf{a}^\delta, \beta)$ be the DRUM model in which $N = 1$, $L = 2$, and $\beta = 0.8$. All non-zero elements of $\mathbf{a}^1, \cdots, \mathbf{a}^\delta$ are $\mathbf{a}^2(1, 1, 1) = 0.5$ and $\mathbf{a}^2(1, 2, 1) = 1$. We show that there exists no Datalog program that is faithful for $\mathcal{M}$ using proof by contradiction.

Assume that $\mathcal{R}$ is a faithful Datalog program for $\mathcal{M}$, i.e., for any dataset $\mathcal{D}$, $T_{\mathcal{M}}(\mathcal{D}) = T_{\mathcal{R}}(\mathcal{D})$. Consider two datasets $\mathcal{D}_1 = \{R_1(c_1, c_1), R_2(c_3, c_3)\}$ and $\mathcal{D}_2 = \mathcal{D}_1 \cup \{R_1(c_1, c_2), R_1(c_2, c_2)\}$, where constant $c_2$ does not appear in $\mathcal{R}$. According to Equation (3), the model $\mathcal{M}$ predicts no fact if it is applied to $\mathcal{D}_1$ (i.e., $T_{\mathcal{M}}(\mathcal{D}_1) = \emptyset$) while it predicts the fact $R_2(c_1, c_2)$ if it is applied to $\mathcal{D}_2$ (i.e., $T_{\mathcal{M}}(\mathcal{D}_2) = \{R_2(c_1, c_2)\}$). On the other hand, since $\mathcal{R}$ is complete for $\mathcal{M}$, we have $R_2(c_1, c_2) \in T_{\mathcal{R}}(\mathcal{D}_2)$. Let $\sigma$ be a mapping from constants to constants with $\sigma(c_2) = c_1$ and $\sigma(c_i) = c_i$ for any $i \neq 2$. Observe that $\sigma(\mathcal{D}_2) = \mathcal{D}_1$. Since $\sigma(R_2(c_1, c_2)) = R_2(c_1, c_1)$, we have $R_2(c_1, c_2) \in \sigma(T_{\mathcal{R}}(\mathcal{D}_2))$. It a well-known property of Datalog that for each Datalog program $\mathcal{R}'$ and each substitution $\mu$ that maps each constant in $\mathcal{R}'$ to itself, it holds that $\mu(T_{\mathcal{R}'}(D)) \subseteq T_{\mathcal{R}'}(\mu(D))$. Since constant $c_2$ does not occur in $\mathcal{R}$ and $\sigma$ is an identity mapping for other constants, we have $\sigma(T_{\mathcal{R}}(\mathcal{D}_2)) \subseteq T_{\mathcal{R}}(\sigma(\mathcal{D}_2)) = T_{\mathcal{R}}(\mathcal{D}_1)$, indicating $R_2(c_1, c_2) \in T_{\mathcal{R}}(\mathcal{D}_1)$. Notice that $T_{\mathcal{R}}(\mathcal{D}_1) \not\subseteq T_{\mathcal{M}}(\mathcal{D}_1)$, so $\mathcal{R}$ is not sound for $\mathcal{M}$, which contradicts our assumption that $\mathcal{R}$ is faithful for $\mathcal{M}$. Thus, no faithful Datalog program exists for $\mathcal{M}$. $\qquad\square$

**Theorem 3.** *Program* $\mathcal{R}_{\mathcal{M},\gamma}^{\text{MP}}$ *extracted by Algorithm 1 on input* $\mathcal{M} = (\mathbf{a}^1, \cdots, \mathbf{a}^\delta, \beta)$ *is faithful to* $\mathcal{M}$ *for* $\gamma = \beta$. *Furthermore, Algorithm 1 terminates in* $\mathcal{O}\left(L \cdot (2\delta)^{L+1} \left(N + \omega^{(2\,\delta)^{L+1}}\right)\right)$ *steps, where* $\omega$ *is the maximum value of* $\omega(r)$ *for* $r$ *a rule of the form* (2) *with* $\ell \leq L$.

*Proof.* Let $\mathcal{R}_{\mathcal{M},\beta}^{\text{MP}}$ be the output of the algorithm for $\mathcal{M}$ with $\gamma = \beta$. Let $L$ and $N$ be the depth and rank of $\mathcal{M}$, respectively.

**(Auxiliary Claim)** We first prove an auxiliary claim: for each rule $r \in \mathcal{R}_{h,L}^{\text{PATH}}$ with head atom $R_h(x, y)$, the value $\varphi_r$ is defined after line 9 by Algorithm 1 during the iteration of index $h$ (line 3). Besides, $\varphi_r = \varphi_{\mathcal{M}}(r)$. Let $\psi_{k_1} \wedge \cdots \wedge \psi_{k_\ell}$ be the body atoms of $r$, where $1 \leq k_i \leq \delta$ for each $k_i$, and each $\psi_{k_i}$ is of the form $R_{k_i}(z_{i-1}, z_i)$ or $R_{k_i}(z_i, z_{i-1})$. Note that $\varphi_r$ is only created and modified within the loop of $h$ (line 3). Moreover, lines 4–10 ensure that $\varphi_r = \sum_{(k_1', \cdots, k_L') \in \mathcal{S}} \sum_{i=1}^N \prod_{j=1}^L \mathbf{a}^h(i, j, k_j')$, where $\mathcal{S}$ is the set of all sequences $(k_1', \cdots, k_L')$ with $1 \leq k_i' \leq 2\delta+1$. Therefore, when performing the iteration in lines 6–7 with index $[k_1', \cdots, k_L']$, rule $r$ is produced in line 8, and so $\varphi_r$ is defined. To complete the proof, by Equation (11) we need to show that $\mathcal{S} = \mathcal{S}_r^L$. This, however, follows straightforwardly from the definitions of $\mathcal{S}$ and $\mathcal{S}_r^L$.

**(Soundness)** We prove the soundness of $\mathcal{R}_{\mathcal{M},\beta}^{\text{MP}}$ for $\mathcal{M}$ by considering an arbitrary dataset $\mathcal{D}$ and showing that $T_{\mathcal{R}_{\mathcal{M},\beta}^{\text{MP}}}(\mathcal{D}) \subseteq T_{\mathcal{M}}(D)$. Let $R_h(c_s, c_t)$ be an arbitrary fact in $T_{\mathcal{R}_{\mathcal{M},\beta}^{\text{MP}}}(\mathcal{D})$. Then, there exists a multipath rule $r$ in $\mathcal{R}_{\mathcal{M},\beta}^{\text{MP}}$ such that either (case 1) the body of $r$ is $\top$, or (case 2) the body of $r$ is of the form $\phi_1 \wedge \cdots \wedge \phi_P$ for $P \geq 1$, with each $\phi_p$ being a multipath conjunction with a distinct core (note that Algorithm 1 never outputs rules containing two multipath conjunctions with the same core (line 15–16)), and there exists a substitution $\sigma$ from the variables in $r$ to constants of $\mathcal{D}$ that grounds $x$ to $c_s$ and $y$ to $c_t$ such that, for each $\phi_p$ of the form (7), if its core is $\Psi^p = \psi_{k_1}^p \wedge \cdots \wedge \psi_{k_{\ell_p}}^p$ and its cardinality is $C_p$, then $\psi_{k_i}^p \sigma \in \mathcal{D}$ for each $1 \leq i \leq \ell_p$, and for each two $1 \leq j < j' \leq C_p$, there exists $1 \leq i \leq \ell_p$ such that $z_i^j \sigma \neq z_i^{j'} \sigma$.

*Case 1.* If the body of $r$ is $\top$, then $r = r_{|\Omega|}$. The auxiliary claim shows that $\varphi_{r_{|\Omega|}} = \varphi_{\mathcal{M}}(r)$. By Lemma 1, $\mathbf{v}_s^h(t) \geq \varphi_{\mathcal{M}}(r)$ since $\varphi_{\mathcal{M}}(r')$ and $\mathbf{q}_{r',\mathcal{D},s}$ are non-negative for each $r' \in \mathcal{R}_{h,L}^{\text{PATH}}$. Thus, $\mathbf{v}_s^h(t) \geq \varphi_{r_{|\Omega|}}$. Meanwhile, $\varphi_{r_{|\Omega|}}$ is precisely the value compared with $\gamma$ in line 18, in the iteration starting from line 14 with index $[C_1, \cdots, C_{|\Omega|}] = [0, \cdots, 0]$. Since $r \in \mathcal{R}_{\mathcal{M},\beta}^{\text{MP}}$, $\varphi_{r_{|\Omega|}} > \beta$. Therefore, $R_h(c_s, c_t) \in T_{\mathcal{R}_{\mathcal{M},\beta}^{\text{MP}}}(\mathcal{D})$.

*Case 2.* For each $1 \leq p \leq P$, if $\varphi_p$ is of the form (7), we can use $\sigma$ to produce $C_p$ substitutions $\sigma_1, \cdots, \sigma_{C_p}$ defined as $\sigma_j(z_i^j) = z_i \sigma$ for each $0 \leq i \leq \ell$ and $1 \leq j \leq C_p$. If $C_p > 1$, then all substitutions are necessarily pairwise different (i.e., they differ in the assignment of at least one variable) because line 13 ensures that $C_p$ can only be greater than 1 if $\Psi^p$ contains at least two body atoms, and so $\ell > 1$. Thus, for each pair of $j$ and $j'$ satisfying $1 \leq j < j' \leq C_p$, there exists $1 \leq i \leq \ell - 1$ such that $z_i^j \sigma \neq z_i^{j'} \sigma$. Therefore, $\mathbf{q}_{r_p, \mathcal{D}, s}(t) \geq C_p$, for $r_p$ as the rule $R_h(x, y) \leftarrow \Psi^p$. By Lemma 1, $\mathbf{v}_s^h(t) = \sum_{r \in \mathcal{R}_{h,L}^{\text{PATH}}} \varphi_{\mathcal{M}}(r) \cdot \mathbf{q}_{r, \mathcal{D}, s}(t)$. By construction, each rule $r_p$ is in $\mathcal{R}_{h,L}^{\text{PATH}}$. Both $\varphi_{\mathcal{M}}(r)$ and $\mathbf{q}_{r, \mathcal{D}, s}$ are non-negative, so $\mathbf{v}_s^h(t) \geq \sum_p \varphi_{\mathcal{M}}(r_p) \cdot C_p$ if $t \neq s$ or $\mathbf{v}_s^h(t) \geq \sum_p \varphi_{\mathcal{M}}(r_p) \cdot C_p + \varphi_{\mathcal{M}}(R_h(x, x) \leftarrow \top)$ otherwise. By the auxiliary claim, this is equal to $\sum_p \varphi_{r_p} \cdot C_p$ or $\sum_p \varphi_{r_p} \cdot C_p + \varphi_{r_{|\Omega|}}$, respectively, which is precisely the value compared with $\gamma$ in line 17 (if $c_s \neq c_t$), or line 18 (if $c_s = c_t$), in the iteration of the loop starting from line 14 with index $[C_1', \cdots, C_{|\Omega|-1}']$, where $C_i' = C_p$ if $\Omega(i) = \Psi^p$ and otherwise $C_i' = 0$. This list is well-defined since no two $p$ and $p'$ with $1 \leq p < p' \leq P$ correspond to the same elements of $\Omega$, as we already observed that each multipath conjunction in the body of $r$ has a different core. Hence, $r \in \mathcal{R}_{\mathcal{M},\beta}^{\text{MP}}$, and we have $\sum_p \varphi_{r_p} \cdot C_p + \varphi_{r_{|\Omega|}} > \beta$. Thus, we have $\mathbf{v}_s^h(t) > \beta$, indicating $R_h(c_s, c_t) \in T_{\mathcal{R}_{\mathcal{M},\beta}^{\text{MP}}}(\mathcal{D})$.

**(Completeness)** To prove completeness, we again consider an arbitrary dataset $\mathcal{D}$ and show that $T_{\mathcal{M}}(D) \subseteq T_{\mathcal{R}_{\mathcal{M},\beta}^{\text{MP}}}(\mathcal{D})$. Let $R_h(c_s, c_t)$ be an arbitrary fact in $T_{\mathcal{M}}(D)$. We show that there is a multipath rule in $T_{\mathcal{R}_{\mathcal{M},\beta}^{\text{MP}}}$ which derives the same fact from $\mathcal{D}$. By Lemma 1, $\mathbf{v}_s^h(t) = \sum_{r \in \mathcal{R}_{h,L}^{\text{PATH}}} \varphi_{\mathcal{M}}(r) \cdot \mathbf{q}_{r, \mathcal{D}, s}(t)$. Let $\mathcal{R}$ be the set of all rules $r \in \mathcal{R}_{h,L}^{\text{PATH}}$ such that $\mathbf{q}_{r, \mathcal{D}, s}(t) > 0$. Let $r$ be an arbitrary such rule. By definition, if the body of $r$ is empty, then $\mathbf{q}_{r, \mathcal{D}, s}(t) = 1$. Otherwise, $\mathbf{q}_{r, \mathcal{D}, s}(t)$ is the number of different substitutions that ground the body of $r$ in $\mathcal{D}$ mapping $x$ to $c_s$ and $y$ to $c_t$. Furthermore, in the latter case, the body of $r$ is an element of $\Omega$ and therefore line 13 ensures that some value $\omega_r$ is computed for it. We now consider two possible cases and show that completeness holds in both. In particular, in each case we construct a rule $r'$ with head $R_h(x, y)$ and then show that, on the one hand, the body of $r'$ can be grounded in $\mathcal{D}$ by a substitution that maps $x$ to $c_s$ and $y$ to $c_t$. On the other hand, we show $r' \in \mathcal{R}_{\mathcal{M},\beta}^{\text{MP}}$, and so $R_h(c_s, c_t) \in T_{\mathcal{R}_{\mathcal{M},\beta}^{\text{MP}}}(\mathcal{D})$.

*Case 1.* There exists $r \in \mathcal{R}$ such that $\mathbf{q}_{r, \mathcal{D}, s}(t) > \omega_r$. This will not happen if the body of $r$ is empty (since $\omega_r$ is not defined for it) or has a single atom, since in that case $\mathbf{q}_{r, \mathcal{D}, s} \leq 1$, but line 13 ensures $\omega_r = 1$. Thus, the body of $r$ has at least two atoms. Consider a multipath rule $r'$ where the head is $R_h(x, y)$ if $c_s \neq c_t$ and otherwise it is $R_h(x, x)$, and its body is a multipath conjunction of cardinality $\omega_r$ and its core is the body of $r$, with $x = y$ if the head does not mention $y$. Since $\mathbf{q}_{r, \mathcal{D}, s}(t) > \omega_r$, there exist at least $\omega_r$ substitutions grounding the body of $r$ in $\mathcal{D}$ mapping $x$ to $c_s$ and $y$ to $c_t$. Since in a multipath conjunction of the form (7), each element $\Psi^j$ shares no variables with the others other than $x$ and $y$, we can take the union of those $\omega_r$ substitutions to obtain a new substitution $\sigma$, and it is clear that this substitution grounds the body of $r'$ in $\mathcal{D}$ mapping $x$ to $c_s$ and $y$ to $c_t$. Now, note that there exists a unique element of $\Omega$ equal to the core of $r'$ (i.e., the body of $r$). Consider the list $[C_1, \cdots, C_{|\Omega|-1}]$ where $C_i = \omega_r$ for the unique $i$ such that $\Omega(i)$ is the body of $r$, and $C_i = 0$ otherwise. Consider the iteration in line 14 during the execution of the algorithm within the loop of $h$, with this list as an index. The value compared with $\gamma$ in line 17 is $\varphi_{r_{|\Omega|}} + \omega_r \cdot \varphi_r \geq \omega_r \cdot \varphi_r$ (by the auxiliary claim and the fact that $\varphi_{\mathcal{M}}$ is non-negative). Since $r$ has at least two body atoms, by line 13 we have $\omega_r \cdot \varphi_r > \beta$, and so $r \in \mathcal{R}_{\mathcal{M},\beta}^{\text{MP}}$.

*Case 2.* There exists no $r \in \mathcal{R}$ such that $\mathbf{q}_{r, \mathcal{D}, s}(t) > \omega_r$. Consider the multipath rule $r'$ where the head is $R_h(x, y)$ if $c_s \neq c_t$ and otherwise it is $R_h(x, x)$, and its body has a multipath conjunction for each element $r \in \mathcal{R}$ with non-empty body, with core equal to the body of $r$ (with $x = y$ if $y$ is not mentioned in the head) and cardinality $\mathbf{q}_{r, \mathcal{D}, s}(t)$. This is well defined by our assumption that $\mathbf{q}_{r, \mathcal{D}, s}(t) > 0$. Therefore, for each $r$ with non-empty body, there exist at least $\mathbf{q}_{r, \mathcal{D}, s}(t)$ different substitutions grounding the body of $r$ in $\mathcal{D}$ mapping $x$ to $c_s$ and $y$ to $c_t$. Since in a multipath conjunction of the form (7), each element $\Psi^j$ shares no variables with the others other than $x$ and $y$, and different multipath conjunctions in the body of $r'$ also share no variables other than $x$ and $y$, we can take the union of all those substitutions to obtain a new substitution $\sigma$, and this substitution clearly grounds the body of $r'$ in $\mathcal{D}$ mapping $x$ to $c_s$ and $y$ to $c_t$. Let $[C_1, C_2, \cdots, C_{|\Omega|} - 1]$ be the list which has $C_i = 0$ if $R_h(x, y) \leftarrow \Omega(i)$ is not in $\mathcal{R}$ and $C_i = \mathbf{q}_{r, \mathcal{D}, s}(t)$ if $\Omega(i)$ is equal to the

body of $r$. This list is well-defined since each rule in $\mathcal{R}$ has a different body and hence there exists a unique element in $\Omega$ equal to its body. This list will be considered in the loop of line 14 during the execution of the algorithm, within the loop of $h$, because $\mathbf{q}_{r,\mathcal{D},s}(t) \leq \omega_r$. The value compared with $\gamma$ in line 17–18 for this iteration will be $\sum_{r \in \mathcal{R}} \mathbf{q}_{r,\mathcal{D},s}(t) \cdot \varphi_r$. By the auxiliary claim, this expression is equal to $\sum_{r \in \mathcal{R}} \varphi_{\mathcal{M}}(r) \cdot \mathbf{q}_{r,\mathcal{D},s}(t)$. By Lemma 1 and the fact that for any $r \in \mathcal{R}_{h,L}^{\text{PATH}} \notin \mathcal{R}$, we have $\mathbf{q}_{r,\mathcal{D},s}(t) = 0$ (with our definition of $\mathcal{R}$), we obtain that $\mathbf{v}_s^h(t) = \sum_{r \in \mathcal{R}} \mathbf{q}_{r,\mathcal{D},s}(t) \cdot \varphi_r$. Hence, since $\mathbf{v}_s^h(t) > \beta$ and our assumption that $R_h(c_s, c_t) \in T_{\mathcal{M}}(D)$, we have that line 17–18 ensures $r' \in \mathcal{R}_{\mathcal{M},\beta}^{\text{MP}}$, as we intended to show.

**(Time Complexity)** In line 2, each conjunction has size at most $L$, so the cost of this step is $\mathcal{O}(|\Omega| \cdot L)$. We next analyze the time complexity of the main loop of the algorithm, which runs $\delta$ times. The loop in line 4–10 considers $(2\delta + 1)^L$ different lists. For each list, line 5 requires $\mathcal{O}(L)$ steps, and so does each iteration in line 6–7. The operation in line 9–10 requires $N \cdot L$ steps, so the total cost of this part is $\mathcal{O}((2\delta + 1)^L \cdot (L + L + 1 + N \cdot L))) = \mathcal{O}((2\delta)^L \cdot N \cdot L)$.

The loop in line 11–13 is performed $|\Omega|$ times, each of which requires a constant number of operations. The overall time cost is $\mathcal{O}(|\Omega|)$.

For the loop in line 14–18, let $\omega = \max_{1 \leq i \leq |\Omega|} \omega_i$. The number of possible combinations of cardinalities $[C_1, \cdots, C_{|\Omega|}]$ with $0 \leq C_i \leq \omega_i$ is bounded by $(\omega + 1)^{|\Omega|}$, which is a bound on the number of loop iterations. In each iteration, line 15–16 writes a rule with at most $|\Omega|$ multipath conjunctions, where each conjunction has at most $L \cdot \omega$ atoms and $\binom{\omega}{2} \cdot L \leq \omega^2 \cdot L$ inequalities. Line 17–18 requires a number of operations linear in $\Omega$. Finally, the filtration in line 17–18 can be finished in constant time. The overall time cost of this loop is

$$\mathcal{O}\left((\omega + 1)^{|\Omega|} \cdot \left(|\Omega| \cdot \left(L \cdot \omega + \binom{\omega}{2} \cdot L\right) + |\Omega|\right)\right),$$

which can be simplified to

$$\mathcal{O}\left(\omega^{|\Omega|+2} \cdot |\Omega| \cdot L\right).$$

Therefore, the overall time complexity of Algorithm 1 is

$$\mathcal{O}\left(|\Omega| \cdot L + \delta\left((2\delta)^L \cdot N \cdot L + |\Omega| + \omega^{|\Omega|+2} \cdot |\Omega| \cdot L\right)\right), \tag{12}$$

Considering that the number of conjunctions in $\Omega$ for each length $0 \leq \ell \leq L$ is $(2\delta)^\ell$, we have

$$|\Omega| = \sum_{\ell=0}^{L} (2\delta)^\ell = \sum_{\ell=1}^{L} (2\delta)^\ell + 1 = \frac{2\delta \cdot \left((2\delta)^L - 1\right)}{2\delta - 1} + 1 = \mathcal{O}\left((2\delta)^L\right).$$

Hence, Expression (12) becomes

$$\mathcal{O}\left(\delta(2\delta)^L \cdot N \cdot L + L \cdot (2\delta)^{L+1} \cdot \omega^{(2\delta)^L+2}\right) = \mathcal{O}\left(L \cdot (2\delta)^{L+1}\left(N + \omega^{(2\delta)^{L+1}}\right)\right).$$

$\square$

**Theorem 4.** *Program $\mathcal{R}_{\mathcal{M}}^{\mathcal{D}}$ extracted by Algorithm 2 for a* DRUM *model $\mathcal{M} = (\mathbf{a}^1, \cdots, \mathbf{a}^\delta, \beta)$ and a dataset $\mathcal{D}$ satisfies $T_{\mathcal{R}_{\mathcal{M}}^{\mathcal{D}}}(\mathcal{D}) = T_{\mathcal{M}}(\mathcal{D})$. Furthermore, Algorithm 2 terminates with time complexity $\mathcal{O}((2\delta)^L \cdot (N \cdot L + 2\delta \cdot \epsilon^{L+2}))$.*

*Proof.* To show $T_{\mathcal{R}_{\mathcal{M}}^{\mathcal{D}}}(\mathcal{D}) \subseteq T_{\mathcal{M}}(\mathcal{D})$, we simply point out that the rules in $\mathcal{R}_{\mathcal{M}}^{\mathcal{D}}$ are in $\mathcal{R}_{\mathcal{M},\beta}^{\text{MP}}$ by construction, and so $T_{\mathcal{R}_{\mathcal{M}}^{\mathcal{D}}}(\mathcal{D}) \subseteq T_{\mathcal{R}_{\mathcal{M},\beta}^{\text{MP}}}(\mathcal{D})$. Meanwhile, by Theorem 3, $\mathcal{R}_{\mathcal{M},\beta}^{\text{MP}}$ is sound for $\mathcal{M}$, so $T_{\mathcal{R}_{\mathcal{M},\beta}^{\text{MP}}}(\mathcal{D}) \subseteq T_{\mathcal{M}}(\mathcal{D})$.

To show $T_{\mathcal{M}}(\mathcal{D}) \subseteq T_{\mathcal{R}_{\mathcal{M}}^{\mathcal{D}}}(\mathcal{D})$, we consider an arbitrary fact $R_h(c_s, c_t) \in T_{\mathcal{M}}(\mathcal{D})$ and then show that the rule produced during the iteration of the algorithm for this fact suffices to derive this fact in $\mathcal{D}$. Let $r$ be the rule added to $\mathcal{R}_{\mathcal{M}}^{\mathcal{D}}$ when $R_h(c_s, c_t)$ is used as index in the iteration of loop 11. For each multipath conjunction in its body with core $\psi$, the existence of $\texttt{count}(\psi)$ different paths from $c_s$ to $c_t$ in $\mathcal{D}$ and the fact that $\texttt{count}(\psi)$ is greater or equal to the cardinality of this multipath conjunction ensure that the body of $r$ matches the dataset mapping $x$ to $c_s$ and $y$ to $c_t$. Hence the rule fires on $\mathcal{D}$, and so $R_h(c_s, c_t) \in T_{\mathcal{R}_{\mathcal{M}}^{\mathcal{D}}}(\mathcal{D})$.

---

**Algorithm 3:** SMDRUM Rule Extraction.

---

1   $\mathcal{R} := \emptyset$;
2   $\Omega :=$ list of all conjunctions in the body of the form (2), with $0 \le \ell \le L$, and with no overlapping
    variables other than $x$ and $y$;
3   **foreach** $h \in \{1, \cdots, \delta\}$ **do**
4      **foreach** $i \in \{1, \cdots, N\}$ **do**
5         **foreach** $[k_1, \cdots, k_L] : 1 \le k_i \le 2\delta+1$ **do**
6            $[k'_1, \cdots, k'_\ell] :=$ remove all $2\delta+1$;
7            **foreach** $j \in \{1, \cdots, \ell\}$ **do**
8               **if** $k'_j \le \delta$ **then** $\psi_j := R_{k'_j}(z_{j-1}, z_j)$; **else** $\psi_j := R_{k'_j - \delta}(z_j, z_{j-1})$;
9            **if** $\ell \ge 1$ **then** $r := R_h(x, y) \leftarrow \bigwedge_{j=1}^{\ell} \psi_j$; **else** $r := R_h(x, x) \leftarrow \top$;
10            **if** $\varphi_{r,i}$ is undefined **then** $\varphi_{r,i} := 0$;
11            $\varphi_{r,i} := \max(\varphi_{r,i}, \prod_{j=1}^{L} \mathbf{a}^h(i, j, k_j))$;
12      **foreach** $[\rho_{e_1}, \cdots, \rho_{e_n}] : 1 \le e_1 < \cdots < e_n \le N, \rho_{e_i} \in \Omega$ **do**
13         **if** $\sum_{i=1}^{n} \varphi_{r_{e_i}, e_i} > \gamma$ **then**
14            **if** $\top \notin \{\rho_{e_1}, \cdots, \rho_{e_n}\}$ **then** $\mathcal{R} := \mathcal{R} \cup \{R_h(x, y) \leftarrow \bigwedge_{i=1}^{n} \rho_{e_i}\}$;
15            **else** $\mathcal{R} := \mathcal{R} \cup \{R_h(x, x) \leftarrow \bigwedge_{i=1}^{n} \rho_{e_i}\{y \mapsto x\}\}$;
16   **return** $\mathcal{R}$;

---

**(Time Complexity)**   In the worst case, the number of possible paths in $\mathcal{P}_{\text{all}}$ reaches $\mathcal{O}(\epsilon^L \cdot (2\delta)^L)$, and the size of $T_{\mathcal{M}}(\mathcal{D})$ reaches $\mathcal{O}(\epsilon^2 \cdot 2\delta)$. Consider the iteration in lines 11–19 of Algorithm 2, each loop of a specific path $[c_s, k_1, \cdots, c_t]$ costs at most $\mathcal{O}(L)$ steps. Besides, to compute all the values $\omega(r)$ for each possible rule $r$ of the form (2) requires at most $\mathcal{O}(N \cdot L \cdot (2\delta)^L)$ steps. Therefore, the worst case time complexity of Algorithm 2 is

$$\mathcal{O}(\epsilon^L \cdot (2\delta)^L \cdot \epsilon^2 \cdot 2\delta + N \cdot L \cdot (2\delta)^L) = \mathcal{O}((2\delta)^L \cdot (N \cdot L + 2\delta \cdot \epsilon^{L+2})).$$

$\square$

**Proposition 1.** *For an* SMDRUM *model* $\mathcal{M}$ *there exists a program* $\mathcal{R}_{\mathcal{M}}^{\text{SM}}$ *of inequality-free multipath rules computable in exponential time in* $\mathcal{M}$*'s size such that* $T_{\mathcal{M}}(\mathcal{D}) = T_{\mathcal{R}_{\mathcal{M}}^{\text{SM}}}(\mathcal{D})$ *for any dataset* $\mathcal{D}$.

To prove Proposition 1, we first present the algorithm extracting the relevant program $\mathcal{R}_{\mathcal{M}}^{\text{SM}}$ from $\mathcal{M}$ and then prove the faithfulness of the program and its complexity. The Proposition is then a trivial corollary from that result.

Algorithm 3 extracts a faithful program from an SMDRUM model. For each possible rule $r$ of the form (2) with head atom $R_h(x, y)$ (line 3), the algorithm first computes a score $\varphi_{r,i}$ similar to the score $\varphi_r$ from Algorithm 1, but computed separately for each sub-model (line 5–11) and using max instead of sum to aggregate different products of elements of $\mathbf{a}^h$. Then, it checks all possible combinations of $n$ rules (which are not necessarily different) with head $R_h(x, y)$, and combines their bodies into a single rule (line 14–15).

The following result states that this algorithm extracts faithful programs from SMDRUM models.

**Theorem 6.** *The Datalog program* $\mathcal{R}_{\mathcal{M}}^{\text{SM}}$ *extracted by Algorithm 3 for a* SMDRUM *model* $\mathcal{M} = (\mathbf{a}^1, \cdots, \mathbf{a}^\delta, \beta)$ *is faithful to* $\mathcal{M}$ *for* $\gamma = \beta$. *Furthermore, Algorithm 3 terminates in* $\mathcal{O}(N \cdot (2\delta)^{L \cdot N+1})$ *steps.*

*Proof.* **(Soundness)** We consider an arbitrary fact $R_h(c_s, c_t) \in T_{\mathcal{R}_{\mathcal{M}, \beta}^{\text{SM}}}(\mathcal{D})$ and show that $R_h(c_s, c_t) \in T_{\mathcal{M}}(\mathcal{D})$. Since $R_h(c_s, c_t) \in T_{\mathcal{R}_{\mathcal{M}, \beta}^{\text{SM}}}(\mathcal{D})$, there exists a rule $r \in \mathcal{R}_{\mathcal{M}, \beta}^{\text{SM}}$ and either (1) it is $R_h(x, x) \leftarrow \top$ or (2) it is of the form $R_h(x, y) \leftarrow \bigwedge_{i=1}^{n} b_i$, or analogously with $x = y$, with each $b_i$ a different element of $\Omega$, and a substitution $\sigma$ grounding the body of this rule in $\mathcal{D}$ mapping $x$ to $c_s$ and $y$ to $c_t$. Furthermore, the fact that $r \in \mathcal{R}_{\mathcal{M}, \beta}^{\text{SM}}$ implies that $\sum_{i=1}^{n} \varphi_{r_i, i} > \beta$, for $r_i = R_h(x, y) \leftarrow b_i$. Simultaneously, we can rewrite the equation from Definition 2 as

$$(\mathbf{v}_s^h)^{\mathsf{T}} = \sum_{i=1}^{N} \left( \max_{\substack{(k_1, \cdots, k_L) \in \{1, \cdots, 2\delta+1\}^L \\ (\mathbf{v}_s)^{\mathsf{T}} \cdot \mathbf{M}_{\mathbf{k_1}} \otimes \cdots \otimes \mathbf{M}_{\mathbf{k_L}} = 1}} \left( \prod_{j=1}^{L} \mathbf{a}^h(i, j, k_j) \right) \right), \tag{13}$$

defining the result of $\max$ as 0 of no sequence $(k_1, \cdots, k_L)$ satisfies the relevant conditions. For any $(k_1, \cdots, k_L) \in \{1, \cdots, 2\delta+1\}^L$, the definition of the matrices $\mathbf{M}_{k_j}$ implies that $\mathbf{M}_{k_1} \otimes \cdots \otimes \mathbf{M}_{k_L}(s, t) = 1$ if either $k_1 = \cdots = k_L = 2\delta + 1$, or there exists a substitution $\sigma$ which grounds the body of $r'$—the unique rule such that $(k_1, \cdots, k_L) \in \mathcal{S}_{r'}^L$—in $\mathcal{D}$ mapping $x$ to $c_s$ and $y$ to $c_t$, or otherwise $\mathbf{M}_{k_1} \otimes \cdots \otimes \mathbf{M}_{k_L}(s, t) = 0$. Hence,

$$(\mathbf{v}_s^h)^\mathsf{T}(t) \geq \sum_{i=1}^n \left( \prod_{j=1}^L \mathbf{a}^h(e_i, j, k_j^i) \right) ,$$

where $(k_1^i, \cdots, k_L^i)$ is the element in $\mathcal{S}_{r_i}^L$, (with $r_i = R_h(x, y) \leftarrow b_i$ if $b_i \neq \top$, and $r_i = R_h(x, x) \leftarrow \top$ otherwise) that was used by the algorithm to produce the final value of $\varphi_{r_{e_i}, e_i}$. But then, $\varphi_{r_{e_i}, e_i} = \prod_{j=1}^L \mathbf{a}^h(e_i, j, k_j^i)$, and since the value compared with $\beta$ by the algorithm is $\sum_{i=1}^n \varphi_{r_{e_i}, e_i}$, we have $(\mathbf{v}_s^h)^\mathsf{T}(t) > \beta$, which implies $R_h(c_s, c_t) \in T_\mathcal{M}(\mathcal{D})$.

**(Completeness)** To show completeness, consider an arbitrary $R_h(c_s, c_t) \in T_\mathcal{M}(\mathcal{D})$. Let $e_1 < \cdots < e_n$ be the indices $1 \leq i \leq N$ for which the sequence below is defined.

$$(k_1^i, \cdots, k_L^i) = \underset{\substack{(k_1, \cdots, k_L) \in \{1, \cdots, 2\delta+1\}^L \\ \mathbf{M}_{k_1} \otimes \cdots \otimes \mathbf{M}_{k_L}(s, t) = 1}}{\arg\max} \left( \prod_{j=1}^L \mathbf{a}^h(i, j, k_j) \right) , \tag{14}$$

Let $b_{e_i}$ be the (unique) conjunction such that $(k_1^{e_i}, \cdots, k_L^{e_i}) \in \mathcal{S}_{r_{e_i}}^L$, with $r_{e_i} = R_h(x, y) \leftarrow b_{e_i}$ if $b_{e_i} \neq \top$, and otherwise $r_{e_i} = \top$. Consider the rule $r = R_h(x, y) \leftarrow \bigwedge_{i=1}^n b_{e_i}$ if none of the $r_{e_i}$ have a body equal to $\top$, and otherwise $r = R_h(x, x) \leftarrow \bigwedge_{i=1}^n b_{e_i} \{y \mapsto x\}$. By construction, $\prod_{j=1}^L \mathbf{a}^h(e_i, j, k_j) = \varphi_{r_{e_i}, e_i}$, so Equation (13) and the choice of $(k_1^{e_i}, \cdots, k_L^{e_i})$ together ensure that $(\mathbf{v}_s^h)^\mathsf{T}(t) = \sum_{i=1}^N \varphi_{r_{e_i}, e_i}$, and since $(\mathbf{v}_s^h)^\mathsf{T}(t) > \beta$, then $r \in \mathcal{R}_{\mathcal{M}, \beta}^{\text{SM}}$. To see that $R_h(c_s, c_t) \in T_r(\mathcal{D})$, note simply that since one of the requirements in the equation above is $\mathbf{M}_{k_1} \otimes \cdots \otimes \mathbf{M}_{k_L}(s, t) = 1$; thus, for any $b_{e_i} \neq \top$, there exists a substitution grounding $b_{e_i}$ in $\mathcal{D}$ mapping $x$ to $c_s$ and $y$ to $c_t$. Since, by definition of $\Omega$, different $b_{e_i}$ share no variables other than $x$ and $y$, we can take the union of these substitutions and obtain a substitution that grounds the body of $r$ and maps $x$ to $c_s$ and $y$ to $c_t$.

**(Time complexity)** In Algorithm 3, there are a total of $\delta$ head atoms (line 2). In each rank $i \in \{1, \cdots, N\}$ (line 4), the number of predicate lists of length $L$ (line 5) is:

$$|\{[k_1, \cdots, k_L] : 1 \leq k_i \leq 2\delta + 1\}| = (2\delta + 1)^L . \tag{15}$$

In each iteration of $[k_1, \cdots, k_L]$, the computation of $\varphi_{r_i, i}$ costs at most $\mathcal{O}(L)$ steps. Then in the combination of bodies $b_i$, the overall number of iteration steps is $O((2\delta + 1)^{L \cdot N})$. The insertion of rules in $\mathcal{R}$ has a time cost of $\mathcal{O}(N)$. Therefore, the overall computational cost is:

$$\delta \cdot \left( N \cdot (2\delta + 1)^L \cdot \mathcal{O}(L) + (2\delta + 1)^{L \cdot N} \cdot \mathcal{O}(N) \right) = \mathcal{O}((2\delta)^{L \cdot N + 1}) . \tag{16}$$

$\square$

**Theorem 5.** *For an* MMDRUM *model* $\mathcal{M}$ *of depth* $L$, *the program containing each rule* $r$ *of the form (2) with* $\ell \leq L$ *or* $R(x, x) \leftarrow \top$ *such that* $\alpha_{\mathcal{M}, r} > \beta$ *is faithful for* $\mathcal{M}$.

*Proof.* Let $\mathcal{R}_{\mathcal{M}, \beta}^{\text{MM}}$ be the program stated in the theorem. To show soundness we consider a fact $R_h(c_s, c_t) \in T_{\mathcal{R}_{\mathcal{M}, \beta}^{\text{MM}}}(\mathcal{D})$ and show that it is in $T_\mathcal{M}(D)$. We know that there exists a Datalog rule in $\mathcal{R}_{\mathcal{M}, \beta}^{\text{MM}}$ in the form of (2) or $R_h(x, x) \leftarrow \top$ and, if the body of this rule is not empty, a substitution $\sigma$ that maps $x$ to $c_s$, $y$ to $c_t$, and grounds the body of the rule in $\mathcal{D}$. Let

$$(k_1', \cdots, k_L') = \underset{[k_1'', \cdots, k_L''] \in \mathcal{S}_r^L}{\arg\max} \max_{1 \leq i \leq N} \left( \prod_{j=1}^L \mathbf{a}^h(i, j, k_j'') \right) .$$

---

**Algorithm 4:** MMDRUM Rule Extraction.

---

1    $\mathcal{R} := \emptyset$;

2    **foreach** $h \in \{1, \cdots, \delta\}$ **and** $i \in \{1, \cdots, N\}$ **do**

3       $\mathcal{P} := \emptyset$;   $\mathcal{P}' := \{(1, [])\}$ ;

4       **foreach** $j \in \{1, \cdots, L\}$ **do**

5          $\mathcal{P} := \mathcal{P}'$;   $\mathcal{P}' := \emptyset$;

6          **foreach** $(s, [p]) \in \mathcal{P}$ **and** $k \in \{1, \cdots, \delta\}$ **do**

7             **if** $s \cdot \mathbf{a}^h(i, j, k) > \gamma$ **then** $\mathcal{P}' = \mathcal{P}' \cup \{(s \cdot \mathbf{a}^h(i, j, k), [p, k])\}$;

8       **foreach** $(s, [k_1, .., k_L]) \in \mathcal{P}'$ *with* $s > \gamma$ **do**

9          $[k'_1, \cdots, k'_\ell] :=$ remove all $2\delta + 1$;

10          **foreach** $j \in \{1, \cdots, \ell\}$ **do**

11             **if** $k'_j \leq \delta$ **then** $\psi_j := R_{k'_j}(z_{j-1}, z_j)$; **else** $\psi_j := R_{k'_j - \delta}(z_j, z_{j-1})$;

12          **if** $\ell \geq 1$ **then** $r := R_h(x, y) \leftarrow \bigwedge_{j=1}^{\ell} \psi_j$; **else** $r := R_h(x, x) \leftarrow \top$;

13          $\mathcal{R} := \mathcal{R} \cup \{r\}$;

14   **return** $\mathcal{R}$;

---

Since this rule is in $\mathcal{R}^{\text{MM}}_{\mathcal{M}, \beta}$, we have $\prod_{j=1}^{L} \mathbf{a}^h(i, j, k'_j) > \beta$. Next, analogously to the case for MMDRUM, we can rewrite the equation that defines $(\mathbf{v}^h_s)^\intercal$ in MMDRUM as

$$(\mathbf{v}^h_s)^\intercal(t) = \max_{1 \leq i \leq N} \left( \max_{\substack{(k_1, \cdots, k_L) \in \{1, \cdots, 2\delta+1\}^L \\ \mathbf{M}_{k_1} \otimes \cdots \otimes \mathbf{M}_{k_L}(s,t)=1}} \left( \prod_{j=1}^{L} \mathbf{a}^h(i, j, k_j) \right) \right). \tag{17}$$

Now, note that if the body of $r$ is not $\top$, the existence of $\sigma$ implies that $\mathbf{M}_{k'_1} \otimes \cdots \otimes \mathbf{M}_{k'_L}(s, t) = 1$. If the body is $\top$, then $c_s = c_t$ and $k'_i = 2\delta + 1$, so $\mathbf{M}_{k'_1} \otimes \cdots \otimes \mathbf{M}_{k'_L}(s, t) = 1$ holds by definition of the MMDRUM model. This means that our sequence $(k'_1, \cdots, k'_L)$ is considered in the $\max$ computations, and so we obtain $(\mathbf{v}^h_s)^\intercal(t) \geq \prod_{j=1}^{L} \mathbf{a}^h(i, j, k'_j)$. Thus, $(\mathbf{v}^h_s)^\intercal(t) > \beta$ and hence $R_h(c_s, c_t) \in T_{\mathcal{M}}(\mathcal{D})$.

To show completeness, we consider a fact $R_h(c_s, c_t) \in T_{\mathcal{M}}(\mathcal{D})$. Let

$$i, (k_1, \cdots, k_L) = \underset{\substack{1 \leq i \leq N \\ [k''_1, \cdots, k''_L] \in \mathcal{S}^L_r \\ \mathbf{M}_{k_1} \otimes \cdots \otimes \mathbf{M}_{k_L}(s,t)=1}}{\arg\max} \left( \prod_{j=1}^{L} \mathbf{a}^h(i, j, k''_j) \right).$$

Clearly, at least one such $i$ and sequence must exist for $R_h(c_s, c_t)$ to be in $T_{\mathcal{M}}(\mathcal{D})$. By Equation (17), we have that $(\mathbf{v}^h_s)^\intercal(t) = \prod_{j=1}^{L} \mathbf{a}^h(i, j, k_j)$. Next, consider the (unique) rule $r$ with $(k_1, \cdots, k_L) \in \mathcal{S}^L_r$. By our choice of $i$ and $(k_1, \cdots, k_L)$, we have $\alpha_{\mathcal{M}, r} = \prod_{j=1}^{L} \mathbf{a}^h(i, j, k_j)$. Since $R_h(c_s, c_t) \in T_{\mathcal{M}}(\mathcal{D})$, we have $(\mathbf{v}^h_s)^\intercal(t) > \beta$ and so $\alpha_{\mathcal{M}, r} > \beta$. Thus, $r \in \mathcal{R}^{\text{MM}}_{\mathcal{M}, \beta}$, as we wanted to show. It remains to see that we can ground the body of $r$ in $\mathcal{D}$ mapping $x$ to $c_s$ and $y$ to $c_t$, but this follows from the definition of $(k_1, \cdots, k_L)$, which has the condition $\mathbf{M}_{k_1} \otimes \cdots \otimes \mathbf{M}_{k_L}(s, t) = 1$. $\qquad \square$

## B    EFFICIENT RULE EXTRACTION FOR MMDRUM

We present an optimised procedure to compute the faithful program $\mathcal{R}^{\text{MM}}_{\mathcal{M}, \beta}$ for an MMDRUM model $\mathcal{M}$. The procedure is laid out in Algorithm 4. It relies on two techniques to optimise rule extraction. On the one hand, instead of first enumerating all rules and then calculating their scores, we interweave the construction of the rules with the computation of the scores. This allows us to take advantage of structure sharing. On the other hand, since the elements of $\mathbf{a}^h$ are all between 0 and 1, computing the value of rules by multiplying different elements will result in a monotonically decreasing manner. Therefore, at each step, we can compare the value computed so far with the threshold, thus pruning the search space.

The following proposition ensures that the output of the algorithm corresponds to the target program.

**Proposition 2.** *The Datalog program $\mathcal{R}$ extracted by Algorithm 4 for an* MMDRUM *model* $\mathcal{M} = (\mathbf{a}^1, \cdots, \mathbf{a}^\delta, \beta)$ *is equal to* $\mathcal{R}^{\mathrm{MM}}_{\mathcal{M}, \beta}$. *Furthermore, Algorithm 4 terminates in* $\mathcal{O}(N \cdot \delta^{L+1})$ *steps.*

Notice that, although the time complexity given in Proposition 2 contains an exponential term $\delta^{L+1}$, the real-world $\delta$ and $L$ would be relatively small. Besides, the pruning in each step by threshold $\gamma$ further improves the computational efficiency. The algorithm usually terminates in a short time in practice, which is also confirmed by the empirical results in Section 6.

*Proof.* We show the double inclusion. First, let $\mathcal{R}^{\mathrm{MM}}_{\mathcal{M}, \beta}$ be the program from Theorem 5. If $r \in \mathcal{R}^{\mathrm{MM}}_{\mathcal{M}, \beta}$ with head predicate $R_h$, then there exists $(k'_1, \cdots, k'_L) \in \mathcal{S}^L_r$ and $i \in \{1, \cdots, N\}$ such that $\alpha_{\mathcal{M}, r} > \beta$. Note that any prefix of factors in the expression will also be greater than $\beta$ since elements of $\mathbf{a}^h(i, j, k)$ are in $[0, 1]$. But then, the execution of the algorithm ensures that for the iteration with $h$ and $i$, the path $(s, k'_1, \cdots, k'_L, t)$ and all its subpaths are considered, and since lines 12-16 generate the rule $r$ from $(k'_1, \cdots, k'_L)$, we have that $r$ is in the output of the algorithm.

Conversely, suppose that $r$ is in the output of the algorithm, with head predicate $R_h$. Then we have that there exist $i$ and $(k_1, \cdots, k_L) \in \mathcal{S}^L_r$ such that in the iteration of line 2 with index $i$, the path $(s, k_1, \cdots, k_L, t)$ is considered and $\prod_{j=1}^L \mathbf{a}^h(i, j, k_j) > \beta$. But then, $\max_{(k'_1, \cdots, k'_L) \in \mathcal{S}^L_r} \max_{1 \le i \le N} \prod_{j=1}^L \mathbf{a}^h(i, j, k'_j) > \beta$, and so the rule $r$ is in $\mathcal{R}^{\mathrm{MM}}_{\mathcal{M}, \beta}$.

To prove the complexity, note that the outer loop iterates $\delta \cdot N$ times. For each of these iterations, the number of rules that can be built is $(2\delta + 1)^L$. The remaining operations can be performed in linear time. Hence the complexity is $\mathcal{O}(\delta \cdot N \cdot (2\delta + 1)^L)$, which can be simplified as $\mathcal{O}(N \cdot \delta^{L+1})$. $\square$

## C   EXPERIMENT DETAILS

This section provides additional details regarding the experimental set-up, including the used benchmarks and the configurations used for training and testing.

### C.1   DATASETS

We used the benchmark datasets for inductive KG completion from Teru et al. (2020). Each dataset consists of train, validation and test sets, where the test set contains the same predicates but distinct constants w.r.t. the train and validation sets. To use them for the KG completion task, we further split each train set and test set into an "incomplete" dataset and a set of positive examples with a $3 : 1$ ratio on a random basis. To evaluate the models prediction on negative examples, we randomly sampled the same number of negative facts as positive examples in the test set, and combined both positive and negative facts in the test process.

Table 4 presents the statistics of each dataset used in our experiments. Apart from the number of facts and predicates, we also computed the in-degree and out-degree of nodes in each dataset by viewing it as a graph, where nodes are constants and edges are predicates. Compared with FB15k-237 and WN18RR, NELL-995 datasets generally have smaller median in-degree and out-degree, indicating that they have relatively simpler graph structure and fewer paths within a given length constraint (e.g., $L = 3$).

Family (Kok & Domingos, 2007) is a dataset of blood relationship between individuals. It contains predicates such as *sister*, *brother*, *mother*, *father*, etc. We reused the split of train, validation and test sets in Sadeghian et al. (2019).

### C.2   SETTINGS

**Model Implementation.**   The original implementation[3] of DRUM by Sadeghian et al. (2019) relied on Python 2.7[4] and Tensorflow 1.13[5], which were relatively outdated and did not support most up-to-date operators (e.g., the max-product of matrices). Therefore, we re-implemented the models

---

[3]`https://github.com/alisadeghian/DRUM`
[4]`https://www.python.org/download/releases/2.7/`
[5]`https://pypi.org/project/tensorflow/1.13.1/`

Table 4: Statistics of datasets.

| | | #Train Facts | #Validation Facts | #Test Facts | #Predicates | In-Degree | | Out-Degree | |
|---|---|---|---|---|---|---|---|---|---|
| | | | | | | median | max | median | max |
| FB15k-237 | V1 | 4,245 | 489 | 2,198 | 180 | 2 | 388 | 3 | 69 |
| | V2 | 9,739 | 1,166 | 4,623 | 200 | 2 | 588 | 3 | 102 |
| | V3 | 17,986 | 2,194 | 8,271 | 215 | 3 | 1,061 | 4 | 139 |
| | V4 | 27,203 | 3,352 | 13,138 | 219 | 3 | 1,289 | 5 | 176 |
| NELL-995 | V1 | 4,687 | 414 | 933 | 14 | 1 | 293 | 1 | 188 |
| | V2 | 8,219 | 922 | 5,062 | 88 | 1 | 251 | 2 | 608 |
| | V3 | 16,393 | 1,851 | 8,857 | 142 | 1 | 524 | 2 | 875 |
| | V4 | 7,546 | 876 | 7,804 | 76 | 1 | 254 | 2 | 545 |
| WN18RR | V1 | 5,410 | 630 | 1,806 | 9 | 2 | 43 | 2 | 17 |
| | V2 | 15,262 | 1,838 | 4,452 | 10 | 2 | 93 | 2 | 21 |
| | V3 | 25,901 | 3,097 | 6,932 | 11 | 2 | 142 | 2 | 87 |
| | V4 | 7,940 | 934 | 13,763 | 9 | 2 | 56 | 2 | 24 |
| Family | | 23,483 | 2,038 | 20,450 | 12 | 5 | 50 | 5 | 50 |

with Python 3.8[6] and PyTorch 2.0.1[7]. For NEURAL-LP, we reused the source codes and settings provided in Tena Cucala et al. (2022b).

**Model Training.** We followed Sadeghian et al. (2019) for all the default settings in training, such as the log-likelihood loss function, Adam optimizer and the use of 10 maximum training epochs. An early stop strategy was adopted for each model based on the prediction loss of the validation set. All experiments were conducted on a Linux workstation with a Xeon E5-2670 CPU. Our data and source codes are available at `https://anonymous.4open.science/r/FaithfulRE`

**Rule Extraction.** We conducted rule extraction for each model and benchmark. In addition to using threshold $\gamma = \beta$ in rule extraction, we further assessed their efficiency by comparing the time cost of rule extraction with different $\gamma$.

For MMDRUM, we used the rule extraction threshold $\gamma = 0.1$ and $0.01$, respectively. For SMDRUM, to avoid redundant body atoms brought by predicates with low scores such as Figure 1 (a), to make rule extraction feasible, and to balance the contribution of each sub-model, we required the score contributed by each sub-model to exceed a lower-bound (i.e., $\gamma'$ in Figure 1 (b)). We evaluated $\gamma' = 0.1$ and $0.01$ for SMDRUM in the experiments. Further, we combined duplicate chains of predicates learned by different sub-models in each SMDRUM rule by adding up their confidence scores and retaining the body atoms only once (as shown in Figure 1 (b)).

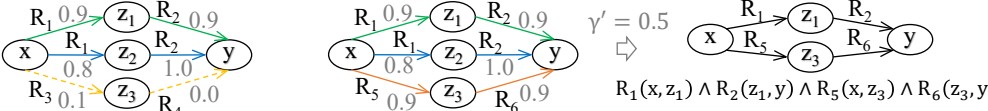

(a) a rule body with redundancy.    (b) filtering and simplifying the rule body of SMDRUM.

Figure 1: A motivating example of SMDRUM rule extraction. Different colors represent ranks. The yellow dotted path in (a) is redundant as it contributes nothing to the overall confidence score of the rule, thus should be removed. (b) shows the filtering and simplifying process of combining body atoms by the same chain of predicates (the green and blue paths) learned from different ranks.

For rule extraction in DRUM, as discussed in Section 4, the relatively high complexity of Algorithm 1 makes it impractical to extract all multipath rules for a DRUM model. Instead, we used the approach from Section 5.1, extracting rules that explain the predictions on the test dataset.

---

[6] `https://www.python.org/downloads/release/python-3817/`
[7] `https://pytorch.org/get-started/pytorch-2.0/`

# D  ADDITIONAL EVALUATION RESULTS

We provide more evaluation results below, including the training and rule extraction time of each model on the datasets, along with analyses of the extracted rules.

## D.1  TRAINING TIME

Table 5 presents the training time of MMDRUM, SMDRUM and DRUM on the FB15k-237, NELL-995, and WN18RR datasets. The three models used similar time for training, which generally increased in accordance with the number of constants and predicates in the dataset. All the models finished in a few hours for each dataset. Besides, the training time increased with the model depth $L$, as the number of learnable parameters (i.e., the elements of $\mathbf{a}^1, \cdots, \mathbf{a}^\delta$) proportionally increased with $L$.

Table 5: Training Time (minutes) of each model with $N = 3$.

| | | MMDRUM | | SMDRUM | | DRUM | |
|---|---|---|---|---|---|---|---|
| | | $L = 2$ | $L = 3$ | $L = 2$ | $L = 3$ | $L = 2$ | $L = 3$ |
| FB15k-237 | V1 | 18.7 | 28.1 | 17.9 | 26.0 | 16.7 | 25.4 |
| | V2 | 69.8 | 119.5 | 64.6 | 99.3 | 62.0 | 93.4 |
| | V3 | 189.5 | 322.4 | 204.9 | 332.9 | 213.2 | 347.9 |
| | V4 | 389.1 | 598.7 | 425.1 | 653.6 | 434.9 | 668.5 |
| NELL-995 | V1 | 1.8 | 3.0 | 2.1 | 3.3 | 1.8 | 2.8 |
| | V2 | 24.9 | 40.3 | 28.1 | 46.1 | 27.2 | 43.6 |
| | V3 | 153.2 | 250.5 | 177.9 | 273.4 | 176.8 | 277.4 |
| | V4 | 21.5 | 35.0 | 24.7 | 40.1 | 23.6 | 37.1 |
| WN18RR | V1 | 1.5 | 2.4 | 1.7 | 2.7 | 1.7 | 3.0 |
| | V2 | 11.5 | 14.9 | 14.1 | 21.3 | 13.3 | 21.1 |
| | V3 | 40.2 | 61.9 | 40.1 | 76.3 | 46.2 | 68.4 |
| | V4 | 7.3 | 12.4 | 9.3 | 14.3 | 6.9 | 9.5 |

## D.2  RULE EXTRACTION TIME

Table 6 shows the rule extraction time of each model for each dataset. Generally, all of them finished extraction in a few minutes for all the datasets, thereby demonstrating their practical viability. Besides, a lower threshold $\gamma$ (or $\gamma'$) leads to an increase of rule extraction time, which is closely related to the increasing number of rules being extracted under $\gamma$ (or $\gamma'$).

Not surprisingly, MMDRUM finished extracting the faithful Datalog program within the shortest time, which is in accordance to its acceptable time complexity $O(N \cdot \delta^{L+1})$. SMDRUM was run for a similar amount of time in best-effort mode. We in fact attempted to implement our faithful Datalog rule extraction for SMDRUM, Algorithm 3. However, with $N = 3$ and $L = 3$ the extraction process had not finished in 3 hours for dataset FB15k-237 V1, which suggested that extracting a faithful program for SMDRUM may not be feasible in practice.

For DRUM, the time cost of rule extraction (using the dataset-dependent Algorithm 2) was very influenced by the actual structure of the dataset. Observe that even though the numbers of predicted facts in different datasets were close, the NELL-995-V1 dataset required much longer time than the others (about 170 seconds for $\gamma = 0.01$ and 120 seconds for $\gamma = 0.1$). This was because the dataset contained some central constants with a fairly high in-degree or out-degree, thus leading to a large amount of paths with length $\leq L$ (at most $1,544,589$ with $L = 3$ in NELL-995-V1).

## D.3  COMPARISON OF RULE EXTRACTION UNDER DIFFERENT RANKS

We extended our analysis to assess the impact of rank $N$ and depth $L$ of the model to the derived rules. To facilitate the investigation, we contrasted the extracted rules from MMDRUM across varying configurations of $N$ and $L$.

Table 6: Rule Extraction Time (seconds) of DRUM, SMDRUM, and MMDRUM.

| | | MMDRUM | | SMDRUM | | DRUM | |
|---|---|---|---|---|---|---|---|
| | | $\gamma = 0.1$ | $\gamma = 0.01$ | $\gamma' = 0.1$ | $\gamma' = 0.01$ | $\gamma = 0.1$ | $\gamma = 0.01$ |
| FB15k-237 | V1 | 5.048 | 5.867 | 7.212 | 8.629 | 0.022 | 0.024 |
| | V2 | 6.824 | 8.084 | 6.827 | 8.593 | 0.101 | 0.132 |
| | V3 | 8.477 | 9.836 | 8.021 | 10.205 | 0.673 | 1.002 |
| | V4 | 8.902 | 9.844 | 8.626 | 10.356 | 3.924 | 4.923 |
| NELL-995 | V1 | 0.005 | 0.011 | 0.005 | 0.069 | 118.971 | 169.851 |
| | V2 | 0.637 | 0.832 | 0.621 | 1.049 | 0.164 | 1.212 |
| | V3 | 2.532 | 2.748 | 2.724 | 3.737 | 11.748 | 16.803 |
| | V4 | 0.435 | 0.586 | 0.412 | 0.749 | 12.851 | 20.683 |
| WN18RR | V1 | 0.002 | 0.004 | 0.002 | 0.043 | 0.029 | 0.035 |
| | V2 | 0.002 | 0.005 | 0.002 | 0.048 | 0.046 | 0.049 |
| | V3 | 0.002 | 0.004 | 0.003 | 0.017 | 0.029 | 0.140 |
| | V4 | 0.002 | 0.003 | 0.002 | 0.008 | 0.200 | 0.233 |

Table 7 presents the highest ten rules along with their associated confidence scores for $N = 1, 2$ and $L = 2, 3$, respectively. For both $L = 2$ and $L = 3$, the algorithm extracted rules with length explicitly less than $L$, such as husband$(x, y) \leftarrow$ wife$(y, x)$, which was in alignment with the analyses in Section 2. Regarding the rank $N$, the results suggested that a MMDRUM model with a greater rank can derive the same rules with higher confidence scores, due to the model acquiring the same rule independently in different sub-models.

Table 7: Rules extracted from Family (Kok & Domingos, 2007) dataset by MMDRUM.

| | | $L = 2$ | | $L = 3$ | |
|---|---|---|---|---|---|
| | | Top 10 rules | Score | Top 10 rules | Score |
| $N = 2$ | | brother$(x,y) \leftarrow$ brother$(x,z) \wedge$ brother$(z,y)$ | 0.999 | sister$(x,y) \leftarrow$ sister$(x,z) \wedge$ sister$(z,y)$ | 0.995 |
| | | sister$(x,y) \leftarrow$ sister$(x,z) \wedge$ sister$(z,y)$ | 0.999 | brother$(x,y) \leftarrow$ brother$(x,z) \wedge$ brother$(z,y)$ | 0.994 |
| | | niece$(x,y) \leftarrow$ sister$(x,z) \wedge$ niece$(z,y)$ | 0.998 | uncle$(x,y) \leftarrow$ brother$(x,z_1) \wedge$ mother$(z_1,z_2) \wedge$ mother$(z_2,y)$ | 0.985 |
| | | uncle$(x,y) \leftarrow$ brother$(x,z) \wedge$ uncle$(z,y)$ | 0.998 | daughter$(x,y) \leftarrow$ sister$(x,z) \wedge$ daughter$(z,y)$ | 0.984 |
| | | son$(x,y) \leftarrow$ brother$(x,z) \wedge$ son$(z,y)$ | 0.994 | sister$(x,y) \leftarrow$ sister$(x,z_1) \wedge$ sister$(z_1,z_2) \wedge$ sister$(z_2,y)$ | 0.983 |
| | | daughter$(x,y) \leftarrow$ sister$(x,z) \wedge$ daughter$(z,y)$ | 0.991 | nephew$(x,y) \leftarrow$ nephew$(x,z) \wedge$ brother$(z,y)$ | 0.978 |
| | | aunt$(x,y) \leftarrow$ sister$(x,z) \wedge$ aunt$(z,y)$ | 0.991 | husband$(x,y) \leftarrow$ wife$(y,x)$ | 0.972 |
| | | sister$(x,y) \leftarrow$ daughter$(x,z) \wedge$ mother$(z,y)$ | 0.991 | aunt$(x,y) \leftarrow$ sister$(x,z_1) \wedge$ sister$(z_1,z_2) \wedge$ nephew$(y,z_2)$ | 0.932 |
| | | nephew$(x,y) \leftarrow$ brother$(x,z) \wedge$ niece$(z,y)$ | 0.990 | nephew$(x,y) \leftarrow$ son$(x,z) \wedge$ brother$(z,y)$ | 0.922 |
| | | aunt$(x,y) \leftarrow$ sister$(x,z) \wedge$ mother$(z,y)$ | 0.980 | niece$(x,y) \leftarrow$ sister$(x,z_1) \wedge$ sister$(z_1,z_2) \wedge$ uncle$(y,z_2)$ | 0.901 |
| $N = 1$ | | sister$(x,y) \leftarrow$ sister$(x,z) \wedge$ sister$(z,y)$ | 0.998 | sister$(x,y) \leftarrow$ sister$(x,z) \wedge$ sister$(z,y)$ | 0.994 |
| | | brother$(x,y) \leftarrow$ brother$(x,z) \wedge$ brother$(z,y)$ | 0.990 | brother$(x,y) \leftarrow$ brother$(x,z) \wedge$ brother$(z,y)$ | 0.988 |
| | | niece$(x,y) \leftarrow$ sister$(x,z) \wedge$ niece$(z,y)$ | 0.989 | nephew$(x,y) \leftarrow$ brother$(x,z) \wedge$ nephew$(z,y)$ | 0.968 |
| | | son$(x,y) \leftarrow$ brother$(x,z) \wedge$ son$(z,y)$ | 0.987 | wife$(x,y) \leftarrow$ husband$(y,x)$ | 0.963 |
| | | nephew$(x,y) \leftarrow$ brother$(x,z) \wedge$ nephew$(z,y)$ | 0.976 | daughter$(x,y) \leftarrow$ sister$(x,z) \wedge$ daughter$(z,y)$ | 0.943 |
| | | aunt$(x,y) \leftarrow$ sister$(x,z) \wedge$ aunt$(z,y)$ | 0.974 | husband$(x,y) \leftarrow$ wife$(y,x)$ | 0.922 |
| | | daughter$(x,y) \leftarrow$ sister$(x,z) \wedge$ daughter$(z,y)$ | 0.970 | aunt$(x,y) \leftarrow$ sister$(x,z_1) \wedge$ sister$(z_1,z_2) \wedge$ niece$(y,z_2)$ | 0.865 |
| | | uncle$(x,y) \leftarrow$ brother$(x,z) \wedge$ son$(y,z)$ | 0.917 | niece$(x,y) \leftarrow$ sister$(x,z_1) \wedge$ sister$(z_1,z_2) \wedge$ uncle$(y,z_2)$ | 0.847 |
| | | father$(x,y) \leftarrow$ husband$(x,z) \wedge$ daughter$(y,z)$ | 0.822 | son$(x,y) \leftarrow$ brother$(x,z) \wedge$ father$(y,z)$ | 0.747 |
| | | wife$(x,y) \leftarrow$ husband$(y,x)$ | 0.795 | uncle$(x,y) \leftarrow$ brother$(x,z_1) \wedge$ brother$(z_1,z_2) \wedge$ nephew$(y,z_2)$ | 0.706 |

## D.4 ADDITIONAL EXAMPLES OF EXTRACTED RULES

We also evaluated the rule extraction performance of each model on other benchmark datasets from Teru et al. (2020). Table 8 shows example rules extracted by Algorithm 4 from the FB15k-237[8], NELL-995 and WN18RR datasets with $N = 3$ and $L = 2$. Most rules are correct and understandable. Many of them have lengths strictly smaller than $L$, which is in accordance with the model design of MMDRUM.

Additionally, we evaluate the rules extracted by Algorithm 3 on the FB15k-237, NELL-995 and WN18RR datasets with $N = 3$ and $L = 2$. Similarly to the results in Section 6 on the Family

---

[8]For conciseness, we represent the predicates in FB15k-237 with their last parts separated by slash. For example, we write "location" for "/people/person/places_lived./people/place_lived/location"

Table 8: Rules Extracted by MMDRUM from Teru et al. (2020).

| Datasets | Example Rules |
|---|---|
| FB15k-237 | $\text{awardNominee}(x, y) \leftarrow \text{awardWinner}(x, y)$
$\text{location}(x, y) \leftarrow \text{placeOfBirth}(x, y)$
$\text{filmProducedBy}(x, y) \leftarrow \text{awardWinner}(x, y)$
$\text{tvProgram}(x, y) \leftarrow \text{programCreator}(y, x)$
$\text{langaugeSpokenIn}(x, y) \leftarrow \text{officialLanguage}(z, x) \wedge \text{serviceLocation}(y, z)$ |
| NELL-995 | $\text{subpartOf}(x, y) \leftarrow \text{agentBelongsToOrganization}(x, y)$
$\text{worksFor}(x, y) \leftarrow \text{organizationHiredPerson}(y, x)$
$\text{teamPlaysAgainstTeam}(x, y) \leftarrow \text{teamPlaysAgainstTeam}(y, x)$
$\text{headQuarteredIn}(x, y) \leftarrow \text{organizationHeadQuarteredInCity}(x, y)$
$\text{personBornInCity}(x, y) \leftarrow \text{personBornInLocation}(x, y)$ |
| WN18RR | $\text{derivationallyRelatedForm}(x, y) \leftarrow \text{derivationallyRelatedForm}(y, x)$
$\text{verbGroup}(x, y) \leftarrow \text{verbGroup}(y, x)$
$\text{hasPart}(x, y) \leftarrow \text{hasPart}(x, z) \wedge \text{hasPart}(z, y)$
$\text{alsoSee}(x, y) \leftarrow \text{alsoSee}(y, x)$
$\text{subsetDomainTopicOf}(x, y) \leftarrow \text{derivationallyRelatedForm}(x, y)$ |

dataset Kok & Domingos (2007), while some rules in the form of (2) can be both extracted by SMDRUM and MMDRUM, Algorithm 3 can also extract rules that are not in the form (2), as shown in Table 9. Many rules tend to have lengths smaller than $L$ and focus on the same sequence of predicates among multiple sub-models of SMDRUM.

Table 9: Rules Extracted by SMDRUM from Teru et al. (2020).

| Datasets | Example Rules |
|---|---|
| FB15k-237 | $\text{film}(x, y) \leftarrow \text{filmSetsDesigned}(x, y) \wedge \text{editedBy}(y, x)$
$\text{film}(x, y) \leftarrow \text{writtenBy}(y, x) \wedge \text{editedBy}(y, x)$
$\text{nominatedFor}(x, y) \leftarrow \text{film}(x, y) \wedge \text{editedBy}(y, x)$ |
| NELL-995 | $\text{worksFor}(x, y) \leftarrow \text{personLeadsOrganization}(x, y) \wedge \text{organizationHiredPerson}(y, x)$
$\text{worksFor}(x, y) \leftarrow \text{topMemberOfOrganization}(x, y) \wedge \text{organizationHiredPerson}(y, x)$
$\text{subpartOf}(x, y) \leftarrow \text{cityCapitalOfCountry}(x, y) \wedge \text{proxyFor}(x, y)$ |
| WN18RR | $\text{subsetDomainTopicOf}(x, y) \leftarrow \text{derivationallyRelatedForm}(x, y) \wedge \text{similarTo}(y, x)$
$\text{alsoSee}(x, y) \leftarrow \text{alsoSee}(y, x) \wedge \text{verbGroup}(x, y)$
$\text{similarTo}(x, y) \leftarrow \text{similarTo}(y, x) \wedge \text{verbGroup}(x, y)$ |

