# OpenReview forum: "Faithful Rule Extraction for Differentiable Rule Learning Models"
_ICLR.cc/2024/Conference — ICLR 2024 poster_

### Official Review · Reviewer_piEQ · 2023-10-25

**Soundness:** 2 fair
**Presentation:** 2 fair
**Contribution:** 3 good
**Rating:** 6
**Confidence:** 3

**Summary:**

In this paper, the authors proposed a solution on how to extract rules from the DRUM model that are faithful, i.e., both complete and sound. The authors provided extensive analysis on the correctness of the proposed algorithm and demonstrate the benefits of their algorithm with some experiments.

**Strengths:**

+ The rule learning problem, in particular, the problem of how to obtain a high-quality set of rules, that the authors are exploring is a critical problem.
+ The authors present a detailed theoretical analysis of how to derive faithful rules from DRUM model, which is quite impressive

**Weaknesses:**

+ I have some concerns about the motivation of the proposed method. It is not clear to me why faithfulness is an important metric for rule learning. What kind of benefits can we obtain for the rule learning problem if the rules can respect faithfulness, e.g., say better generalizability? The authors may need more effort to justify this. One motivating example might be helpful for readers to understand this.
+ I also feel that the proposed solution is too specific. It is only applicable to the DRUM model rather than the general rule learning model, which may limit the applicability of the proposed method in general settings.
+ The overall presentation may need to be improved. I feel that the authors have a lot of technical content to present. However, putting all of them together without appropriate illustrations makes it hard for readers to digest. One suggestion would be using some running examples to intuitively explain the meaning of the notations and equations rather than just leaving them in the paper. Even the prior work such as DRUM paper or Neural ILP paper, there are a lot of visualizations that help users understand the intuitions of the proposed method.
+ I feel that the experimental evaluations are also concerning. The authors only focus on evaluating the performance of the proposed methods. However, no appropriate evaluations (except some simple case studies) are conducted for the faithfulness part, e.g., how many rules from the vanilla DRUM model violate the faithfulness criterion and what is the bad effect of including those rules?

**Questions:**

See above.

---

> ### Author Response · Authors · 2023-11-15
> **Response to Reviewer piEQ**
>
> We thank the reviewer for the valuable comments.
>
> > It is not clear to me why faithfulness is an important metric for rule learning. What kind of benefits can we obtain for the rule learning problem if the rules can respect faithfulness, e.g., say better generalizability? The authors may need more effort to justify this. One motivating example might be helpful for readers to understand this. ...One suggestion would be using some running examples to intuitively explain the meaning of the notations and equations rather than just leaving them in the paper.
>
> As requested, we have incorporated an example in the introduction, highlighting the benefits of ensuring faithfulness. We now also refer to this running example in other parts of the paper. Given the tight space restrictions and the need to rigorously present our technical results, this description is necessarily concise. Nevertheless, we anticipate that it will enhance the overall readability of the paper.
>
>
> > I also feel that the proposed solution is too specific. It is only applicable to the DRUM model rather than the general rule learning model, which may limit the applicability of the proposed method in general settings.
>
> While it is necessary to independently analyse the faithfulness of various rule-learning models, we expect that the strategies employed in our approach will contribute to a broader understanding of the faithfulness of rule-learning models. Indeed, we anticipate that these strategies can serve as a foundation for the future analysis of other models.
>
>
> > The authors only focus on evaluating the performance of the proposed methods. However, no appropriate evaluations (except some simple case studies) are conducted for the faithfulness part, e.g., how many rules from the vanilla DRUM model violate the faithfulness criterion and what is the bad effect of including those rules?
>
> Please note that in Section 6 we perform an extensive analysis of the faithfulness of the rules extracted from the original DRUM model (results are summarised in Table 2). The soundness of the extracted rules is guaranteed by Theorem 1. However, Table 2 shows that
> the rules are not complete, and in fact they only produce a very small fraction (less than 7\%) of the facts predicted by the DRUM model. Therefore, *none* of the rule sets extracted from the original DRUM model is faithful.

---

> > ### Comment · Reviewer_piEQ · 2023-11-18
> > **Thanks for your response**
> >
> > Thanks for the authors' efforts in providing detailed responses! One thing that I am still worried about is that the definitions of completeness and faithfulness are still unclear to me. I know that the authors referred to Cucala et al. (2022b) about such definitions. But without formalization of such important concepts in the paper, the paper seems not to be self-contained. Although the authors mentioned that those definitions are provided in Section 2, I couldn't find them at all...
> >
> > This is also related to my earlier concerns about the lack of evaluations on faithfulness in the experiments. Since no formal definitions of faithfulness and completeness are given and no formulas are provided to show how they are calculated in the experiments, it is unclear to readers to understand how those numbers are obtained and how this is related to the notion of faithfulness or completeness.

---

> > > ### Author Response · Authors · 2023-11-18
> > > **Definitions of completeness and faithfulness**
> > >
> > > Many thanks for your comment.
> > >
> > > The definitions of soundness, completeness and faithfulness can be found at the very end of Section 2 within the paragraph entitled "Relations between Models and Programs". The definitions of the operators T_R and T_M (in the case we consider DRUM as a model) are given also in Section 2 in the paragraphs on Datalog and DRUM, respectively.

---

> > > > ### Comment · Reviewer_piEQ · 2023-11-18
> > > > **Thanks for your explanations**
> > > >
> > > > Thanks for your explanations. I found it. I determined to raise my score from 5 to 6.

---

### Official Review · Reviewer_BVyG · 2023-10-29

**Soundness:** 3 good
**Presentation:** 3 good
**Contribution:** 3 good
**Rating:** 8
**Confidence:** 3

**Summary:**

This paper discusses the growing interest in extracting understandable rules from machine learning models that work with knowledge graphs. These models are used for various tasks like knowledge graph completion, node classification, question answering, and recommendation. However, many of these rule extraction methods lack formal guarantees, which can be a problem when applying these rules in critical or legally required situations. The paper focuses on the DRUM model, a variant of the NEURAL-LP model, which has shown good practical performance. The paper explores whether the rules derived from DRUM are sound and complete, meaning they provide reliable results. The authors propose a new algorithm to ensure both soundness and completeness in the extracted rules. This algorithm, while effective, may be less efficient in practice. They also suggest adding constraints to DRUM models to facilitate rule extraction, even if it reduces their expressive power. The paper points out that DRUM and NEURAL-LP models have technical differences, making it necessary to examine the guarantees separately for DRUM.

**Strengths:**

Good paper interesting, with good ideas and good experiments.

**Weaknesses:**

it seems that a lot of citations are missing, please add them and discuss them

https://arxiv.org/pdf/2301.09559.pdf
Learning Interpretable Rules for Scalable Data Representation and Classification
FINRule: Feature Interactive Neural Rule Learning
https://arxiv.org/pdf/2309.09638.pdf

**Questions:**

-

**Details Of Ethics Concerns:**

-

---

> ### Author Response · Authors · 2023-11-15
> **Response to Reviewer BVyG**
>
> We thank the reviewer for the comments and suggestions. We have uploaded a new version of the paper including the references provided by the reviewer.

---

### Official Review · Reviewer_Crqq · 2023-10-31

**Soundness:** 4 excellent
**Presentation:** 2 fair
**Contribution:** 3 good
**Rating:** 8
**Confidence:** 2

**Summary:**

The paper adapts a study that was previously conducted for extracting rules from the Neural-LP model (which is a neural model for reasoning about knowledge graphs) to DRUM, an alternative architecture for the same task. This adaptation is not entirely trivial, as the authors motivate well, so that this can be considered an original contribution. They also show that, in analogy to the previous work, the extracted rules are not necessarily faithful, and can be unsound or incomplete. An expensive alternative is also proposed as a remedy.

**Strengths:**

This paper seems to be well formalized and contain sufficiently interesting improvements over the state-of-the-art. The evaluation is on a standard benchmark task. The introduction is quite accessible, but after that I was quickly lost.

**Weaknesses:**

I am not sufficiently familiar with this area in order to judge whether the contribution of this work is sufficiently deep for publication, whether the shown results are actually correct, or whether the performed evaluation on this set of benchmark sets is a standard procedure in this area.
What I can judge, however, is that the paper is a) very well written in the sense that a stringent formal presentation is chosen, but also b) not written for an audience outside this immediate community, and the authors also do not make any attempt to make their work more accessible. If you do not know any of the prior works, in particular DRUM, you are lost. I could, e.g., easily follow the terse formal description of Datalog, because I am familiar with that. With the similarly formal presentation of DRUM I was completely lost. I now know what parts this system has, but not what they are used for. I know that path-counting is part of the method, but I don't know what paths are counted, and why this has any significance.
I think the paper is a very nice illustration of the fact that shorter explanations are not necessarily more interpretable.

Minor stuff:
The formalization of the paper seems to be very stringent, the references are not (lower case abbreviations such as gnn, all conference names abbreviated (I don't know all of them), weird references to things like ICML volume 227 or ISWC volume 13489, backslashes in URLs, etc.

**Questions:**

I'm not really familiar with this task, but this seems to be quite a heavy machinery for extracting family relations. In Inductive Logic Programming these are only toy examples. Are there no better tasks for showcasing what you can do?

---

> ### Author Response · Authors · 2023-11-15
> **Response to Reviewer Crqq**
>
> We thank the reviewer for the comments and for pointing out typos and other minor infelicities in the references, which we have fixed in the updated version of the paper.
>
>
> > This paper is not written for an audience outside this immediate community, and the authors also do not make any attempt to make their work more accessible. If you do not know any of the prior works, in particular DRUM, you are lost. I could, e.g., easily follow the terse formal description of Datalog, because I am familiar with that. With the similarly formal presentation of DRUM I was completely lost. I now know what parts this system has, but not what they are used for. I know that path-counting is part of the method, but I don't know what paths are counted, and why this has any significance.
>
> To make the paper more accessible, we have uploaded a new version with running examples that illustrate the key ideas and models discussed in the paper. The new examples are introduced in Section 1  and Section 3 (after Lemma 1), respectively, and we expand upon them in the following sections. Given the tight space restrictions and the need to rigorously present our technical results, the discussion of the examples is necessarily concise. Nevertheless, we anticipate that they will enhance the overall readability of the paper.
>
>
> > I'm not really familiar with this task, but this seems to be quite a heavy machinery for extracting family relations. In Inductive Logic Programming these are only toy examples. Are there no better tasks for showcasing what you can do?
>
> We used the Family dataset in Section 6 to illustrate some simple rules extracted by our model which are intuitive and easy to read. We have now included in the Appendix examples of rules extracted for FB15k-237, WN18RR and NELL-995, which are real-world datasets commonly used as KG completion benchmarks.

---

### Official Review · Reviewer_dHjy · 2023-11-01

**Soundness:** 2 fair
**Presentation:** 2 fair
**Contribution:** 2 fair
**Rating:** 5
**Confidence:** 2

**Summary:**

In the abstract, the authors describe the problem that they are solving as  "methods for extracting interpretable rules from ML
models trained to solve a wide range of tasks over knowledge graphs (KGs), such as KG completion, node classification, question answering and recommendation." and talk about how "Many such approaches, however, lack formal guarantees establishing the precise
relationship between the model and the extracted rules, and this lack of assurance becomes especially problematic when the extracted rules are applied in safety critical contexts or to ensure compliance with legal requirements"

While the work described in the paper seems reasonable, this does not quite seem to match the claim in the abstract. As best as I can tell,
the techniques in the paper do not deal with classification / question answering / recommendations. Further, it is not obvious (either from the description of the technique or the experimental results) that they provide assurances around the "precise relationship between the model and the extracted rules" (while there might be soundness, completeness does not seem to result).

**Strengths:**

The results seem interesting for knowledge graph completion problems.

**Weaknesses:**

It was not obvious how this applies for general ML problems, particularly those around question answering. It was not clear how the algorithm might perform for classification/recommendation systems.

**Questions:**

It was not obvious how this applies for general ML problems, particularly those around question answering. It was not clear how the algorithm might perform for classification/recommendation systems.

---

> ### Author Response · Authors · 2023-11-15
> **Response to Reviewer dHjy**
>
> We thank the reviewer for the valuable comments. We answer the questions raised by the reviewer as follows.
>
> > As best as I can tell, the techniques in the paper do not deal with classification / question answering / recommendations.
>
> The techniques presented in the paper address the general problem of realising transformations between KGs. Tasks such as classification, question answering, and recommendation on KGs can be seen as concrete applications of our approach. Question answering over KGs can be viewed as the problem of learning a function that maps the input knowledge graph to a set of facts representing all answers to a given set of questions. Recommendation can be viewed as learning a function that maps a KG describing user-item interactions to a set of facts where each fact represents a recommendation. Finally, node classification can be seen as the problem of learning a function from a KG to a set of facts stating membership of nodes to given classes.
>
>
> > It is not obvious (either from the description of the technique or the experimental results) that they provide assurances around the "precise relationship between the model and the extracted rules" (while there might be soundness, completeness does not seem to result).
>
> Please note that Theorem 3 in Section 4 ensures the faithfulness (i.e., both soundness *and* completeness) of Algorithm 1 for multipath rule extraction from a DRUM model. Furthermore, Theorem 4 in Section 5.1 proves the faithfulness of Algorithm 2 for multipath rule extraction over a fixed dataset. The corresponding proofs are given in the Appendix.

---

> > ### Comment · Reviewer_dHjy · 2023-11-23
> >
> > Thanks for your clarifications.

---

### Meta-Review · Area_Chair_H6Wc · 2023-12-09

**Metareview:**

The paper proposes a way to extract rules from a knowledge graph (KG) that are faithful. This is done by building on DRUM a previous differentiable rule miner for KGs. The authors propose several variants and analyze their faithfulness properties theoretically. Then, they evaluate them on inductive KG completion tasks. The results are encouraging but mixed, while the variants of DRUM can have higher recall, they are less precise and overall less accurate.

The reviewers debated on the significance of the scope of this work but expressed some positive views concerning its theoretical derivations. At the same time they highlighted how the presentation can be too dense and it is not clear why all the description about the DRUM components are useful for. Some concerns were raised about the limited scope of this work and the theoretical evaluation, as it pertains only rules extracted with a very specific differentiable miner, DRUM.

**Justification For Why Not Higher Score:**

The contribution is quite limited in scope and concerns faithful rule mining in the context of a very specific differentiable KG miner: DRUM.

**Justification For Why Not Lower Score:**

The paper is borderline imho and could be rejected given its limited scope and mixed results.

---

### Decision · Program_Chairs · 2024-01-16

Accept (poster)